# Doa10 is a membrane protein retrotranslocase in ER-associated protein degradation

Claudia C Schmidt, Vedran Vasic, Alexander Stein*

Research Group Membrane Protein Biochemistry, Max Planck Institute for Biophysical Chemistry, Göttingen, Germany

**Abstract** In endoplasmic reticulum-associated protein degradation (ERAD), membrane proteins are ubiquitinated, extracted from the membrane, and degraded by the proteasome. The cytosolic ATPase Cdc48 drives extraction by pulling on polyubiquitinated substrates. How hydrophobic transmembrane (TM) segments are moved from the phospholipid bilayer into cytosol, often together with hydrophilic and folded ER luminal protein parts, is not known. Using a reconstituted system with purified proteins from *Saccharomyces cerevisiae*, we show that the ubiquitin ligase Doa10 (Teb-4/MARCH6 in animals) is a retrotranslocase that facilitates membrane protein extraction. A substrate's TM segment interacts with the membrane-embedded domain of Doa10 and then passively moves into the aqueous phase. Luminal substrate segments cross the membrane in an unfolded state. Their unfolding occurs on the luminal side of the membrane by cytoplasmic Cdc48 action. Our results reveal how a membrane-bound retrotranslocase cooperates with the Cdc48 ATPase in membrane protein extraction.

## Introduction

The endoplasmic reticulum (ER) is a major site for protein folding and maturation in the endomembrane system of the eukaryotic cell. A conserved quality control pathway called ER-associated protein degradation (ERAD) removes misfolded, unassembled and mistargeted proteins from the ER into the cytosol where they are degraded by the proteasome (*Christianson and Ye, 2014*; *Mehrtash and Hochstrasser, 2019*; *Ruggiano et al., 2014*). ERAD thus contributes to protein homeostasis. Its malfunction results in ER stress (*Hwang and Qi, 2018*), and it has been linked to several human diseases (*Guerriero and Brodsky, 2012*; *Qi et al., 2017*). ERAD is part of the ubiquitin proteasome system. Studies in the yeast *Saccharomyces cerevisiae* identified two universally conserved membrane-embedded ubiquitin ligases that ubiquitinate ERAD substrates, Hrd1 (SYVN1 in human) (*Bordallo et al., 1998*; *Hampton et al., 1996*; *Kikkert et al., 2004*; *Nadav et al., 2003*) and Doa10 (TEB-4/MARCH6 in animals, SUD-1 in *Arabidopsis thaliana*) (*Doblas et al., 2013*; *Hassink et al., 2005*; *Swanson et al., 2001*). In higher eukaryotes, a larger variety of ubiquitin ligases plays a role in ERAD (*Olzmann et al., 2013*).

Substrates of ERAD can be soluble luminal proteins, or membrane proteins that either need to be moved across or extracted from the ER membrane. This process, termed retrotranslocation or dislocation, requires the AAA protein Cdc48 (VCP or p97 in animals) (*Bays et al., 2001*; *Garza et al., 2009*; *Jarosch et al., 2002*; *Nakatsukasa et al., 2008*; *Rabinovich et al., 2002*; *Ye et al., 2001*). Cdc48 is recruited to substrates by its cofactors Ufd1 and Npl4 which interact with polyubiquitin chains with lysine 48 linkage (*Meyer et al., 2002*; *Ye et al., 2003*). The Cdc48 complex is thought to generate a pulling force that drives extraction of polyubiquitinated proteins from the membrane. This notion is based on biochemical experiments with soluble proteins (*Bodnar and Rapoport, 2017*; *Olszewski et al., 2019*) and recent cryo-EM structures (*Bodnar et al., 2018*; *Cooney et al.,*

*For correspondence:
alexander.stein@mpibpc.mpg.de

Competing interests: The authors declare that no competing interests exist.

**eLife digest** The inside of a cell contains many different compartments called organelles, which are separated by membranes. Each organelle is composed of a unique set of proteins and performs specific roles in the cell. The endoplasmic reticulum, or ER for short, is an organelle where many proteins are produced. Most of these proteins are then released from the cell or sorted to other organelles. The ER has a strict quality control system that ensures any faulty proteins are quickly marked for the cell to destroy. However, the destruction process itself does not happen in the ER, so faulty proteins first need to leave this organelle. This is achieved by a group of proteins known as endoplasmic reticulum-associated protein degradation machinery (or ERAD for short).

To extract a faulty protein from the ER, proteins of the ER and outside the ER cooperate. First, an ERAD protein called Doa10 attaches a small protein tag called ubiquitin to the faulty proteins to mark them for destruction. Then, outside of the ER, a protein called Cdc48 'grabs' the ubiquitin tag and pulls. But that is only part of the story. Many of the proteins made by the ER have tethers that anchor them firmly to the membrane, making them much harder to remove.

To get a better idea of how the extraction works, Schmidt et al. rebuilt the ERAD machinery in a test tube. This involved purifying proteins from yeast and inserting them into artificial membranes, allowing closer study of each part of the process. This revealed that attaching ubiquitin tags to faulty proteins is only one part of Doa10's role; it also participates in the extraction itself. Part of Doa10 resides within the membrane, and this 'membrane-spanning domain' can interact with faulty proteins, loosening their membrane anchors. At the same time, Cdc48 pulls from the outside. This pulling force causes the faulty proteins to unfold, allowing them to pass through the membrane.

Given these findings, the next step is to find out exactly how Doa10 works by looking at its three-dimensional structure. This could have implications not only for the study of ERAD, but of similar quality control processes in other organelles too. A build-up of faulty proteins can cause diseases like neurodegeneration, so understanding how cells remove faulty proteins could help future medical research.

---

*2019*; *Twomey et al., 2019*), which showed that processive threading of a substrate through the central pore of the Cdc48 hexamer under ATP consumption leads to unfolding.

Apart from the Cdc48 ATPase, membrane proteins of the ERAD machinery are thought to contribute to retrotranslocation. For soluble substrates, Hrd1 forms part of a retrotranslocon pore from the ER lumen to the cytosol (*Baldridge and Rapoport, 2016*; *Carvalho et al., 2010*; *Stein et al., 2014*; *Vasic et al., 2020*), but other components such as the Derlin Der1 are also involved, as shown by biochemical data and a recent cryo-EM structure of the Hrd1 complex (*Mehnert et al., 2014*; *Wu et al., 2020*). Less is known about retrotranslocation of membrane proteins. The machinery that mediates this process needs to be quite versatile, because substrates can exhibit different topologies. They may contain one or multiple transmembrane (TM) segments, stretches of hydrophilic amino acids in luminal loops and tightly folded domains. How these structurally and physicochemically diverse elements move across a phospholipid bilayer during the extraction process is not known. Multipass transmembrane proteins such as members of the Derlin family (Der1 and Dfm1 in yeast; Derlin-1,–2 in animals), Hrd1 and Doa10 have been suggested to act as retrotranslocases for membrane proteins (*Carvalho et al., 2010*; *Hampton and Sommer, 2012*; *Lilley and Ploegh, 2004*; *Neal et al., 2018*; *Swanson et al., 2001*; *Ye et al., 2004*).

Another unresolved but linked question regards the folding state of luminal domains during the retrotranslocation process. It is unclear whether luminal domains are moved across the membrane in a folded state, if unfolding occurs prior to retrotranslocation, potentially by a separate ER luminal machinery, or if unfolding is directly coupled to retrotranslocation (*Brodsky, 2012*; *Shi et al., 2019*).

To address these questions, we investigated ERAD mediated by the ubiquitin ligase Doa10 from *S. cerevisiae*. Doa10 is a 150 kDa protein with 14 TM segments (*Kreft et al., 2006*). Its substrates include single- and multi-spanning membrane proteins of the ER and inner nuclear membrane, but also soluble proteins of the cyto- and nucleoplasm (*Ravid et al., 2006*). No completely soluble, luminal substrates of Doa10 have been described. The degrons of Doa10 substrates can be cytoplasmic (*Furth et al., 2011*; *Swanson et al., 2001*), or within the TM region (*Habeck et al., 2015*).

Furthermore, Doa10 regulates sterol metabolism in plants, fungi and animals by degrading squalene monooxygenase (*Doblas et al., 2013*; *Foresti et al., 2013*). Through degradation of mislocalized membrane proteins, Doa10 has a role in maintaining organelle identity (*Dederer et al., 2019*; *Matsumoto et al., 2019*; *Ruggiano et al., 2016*).

Doa10 works in concert with two ubiquitin conjugating enzymes (E2), the tail-anchored membrane protein Ubc6 and the soluble cytoplasmic protein Ubc7 (*Swanson et al., 2001*). Ubc7 is anchored to the ER membrane by Cue1 (*Biederer et al., 1997*). Experiments with soluble cytoplasmic protein fragments showed that Ubc6 and Ubc7 have different roles in the build-up of polyubiquitin chains. Ubc6 initiates ubiquitin chains by transferring the first ubiquitin moiety, whereas Ubc7 extends ubiquitin chains with mainly lysine 48 linkage (*Weber et al., 2016*). Importantly, Ubc6 is itself an unstable protein and degraded in a Doa10-dependent manner (*Swanson et al., 2001*; *Walter et al., 2001*). Similarly, Ubc6 homologues in plants and mammals have been shown to be unstable and are degraded by the proteasome (*Lam et al., 2014*). Interestingly, in case of the *Arabidopsis thaliana* Ubc6 homologue Ubc32, and the mammalian Ube2J1, Hrd1 was identified as the ubiquitin ligase involved (*Burr et al., 2011*; *Chen et al., 2016*).

Here, we developed a reconstituted system with purified proteins to investigate the role of Doa10 in membrane protein retrotranslocation. This system allowed us to mechanistically investigate membrane protein extraction, without relying on indirect read-outs of downstream reactions such as proteasomal degradation. We show that Doa10 is a membrane protein retrotranslocase. Furthermore, we show how Doa10 cooperates with the Cdc48 ATPase in the extraction of proteins with folded luminal domains.

## Results

We reconstituted purified and fluorescently labeled Doa10 and its substrate Ubc6 into separate liposome populations, together with complementary SNARE proteins (*Figure 1A* and *Figure 1—figure supplement 1, A* to F). The tail-anchored (TA) membrane protein Ubc6 was chosen as a model substrate, to limit the number of membrane proteins in our system and thus its complexity. To achieve efficient liposome fusion, we employed previously well-characterized engineered versions of rat SNAREs involved in synaptic vesicle exocytosis (*Cypionka et al., 2009*; *Hernandez et al., 2012*). Indeed, mixing of the two liposome sets led to SNARE-mediated co-reconstitution of Ubc6 and Doa10 (*Figure 1—figure supplement 1G*). This approach ensures that Doa10 and Ubc6 only interact in the phospholipid bilayer, avoiding non-native interactions that can occur when membrane proteins are mixed in the presence of detergents for co-reconstitution. Protease protection experiments showed that Doa10 was reconstituted mostly in the correct orientation (*Figure 1—figure supplement 1H*). For Ubc6, 45% was correctly oriented, another 45% wrong-side out oriented, and a minor fraction not properly membrane inserted (*Figure 1—figure supplement 1, I* to K).

We postulated that retrotranslocase activity of Doa10 facilitates release of a substrate into the aqueous solution, but that such an event should be energetically disfavored unless the membrane-released state was stabilized by chaperones and re-insertion prevented (*Figure 1B*). To test this hypothesis, we incubated Ubc6 liposomes with Get3, a chaperone for TA proteins (*Mateja et al., 2009*) that also interacts with Ubc6 (*Figure 1—figure supplement 2A*). Liposomes were then immobilized to separate soluble and membrane-bound proteins (*Figure 1—figure supplement 2B*). In the presence of Get3, 43 ± 4% Ubc6 was released from liposomes with co-reconstituted Doa10 (*Figure 1C,D* and *Figure 1—figure supplement 2C*). When co-reconstitution of Doa10 with Ubc6 was prevented by inhibiting SNARE-mediated fusion or when Ubc6 liposomes were fused with liposomes lacking Doa10, only 7–9% of Ubc6 were found in the soluble fraction, representing the fraction of Ubc6 sticking to the outside of the liposome surface. In the absence of Get3, or when we used a Get3 mutant defective in TA protein binding (Get3 I193D) (*Mateja et al., 2009*), we observed no, or drastically reduced release, respectively. Nucleotide hydrolysis was not required for Ubc6 release and an ATPase deficient mutant (Get3 D57N) behaved indistinguishably from wild-type (WT) Get3. This suggests that Doa10 allows for passive movement of its substrate Ubc6 out of the membrane.

Release of Ubc6 from the liposome membrane involves movement of the TM anchor and the luminally encapsulated C-terminus across the lipid bilayer. To directly measure exposure of the C-terminus upon retrotranslocation we used a Ubc6 variant labeled with an AlexaFluor488 (A488)

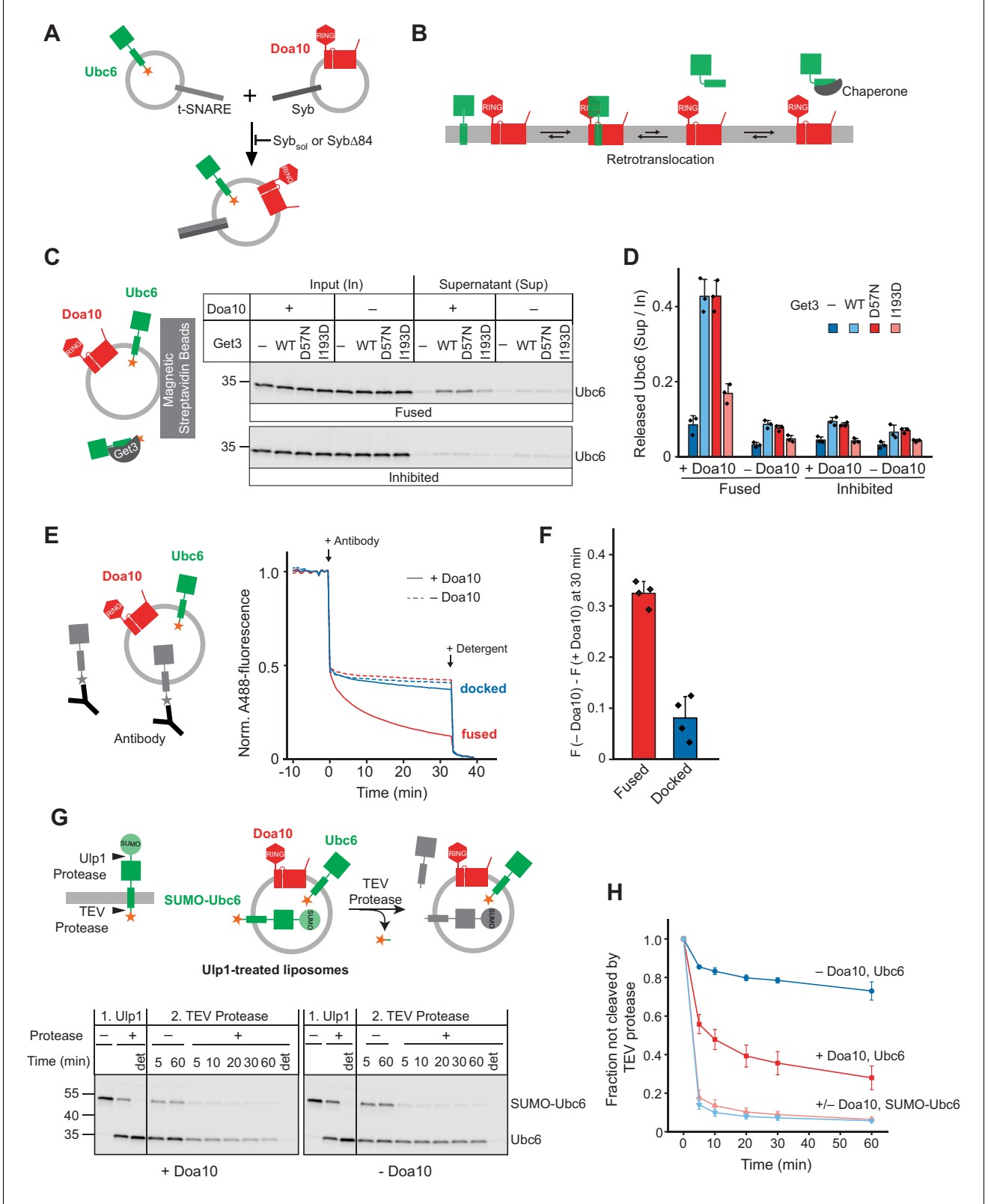

**Figure 1.** Retrotranslocation of Ubc6 by Doa10. (**A**) SNARE-mediated co-reconstitution of Ubc6 and Doa10. Engineered versions of SNAREs involved in synaptic exocytosis were used, that is a Syntaxin 1A fragment, SNAP-25A, and Synaptobrevin 2 (*Pobbati et al., 2006*). Syb$_{sol}$, a cytoplasmic fragment of Synaptobrevin (Syb). SybΔ84, Syb mutant that results in a docked state (*Hernandez et al., 2012*). See *Figure 1—figure supplement 1, E* to K for characterization of liposomes. (**B**) Working hypothesis for retrotranslocation by Doa10. (**C**) Membrane release of Ubc6 in the presence of Get3.

*Figure 1 continued on next page*

*Figure 1 continued*

Fluorescently labeled Ubc6 was co-reconstituted with Doa10 by SNARE-mediated fusion (+ Doa10), as shown in (**A**). Where indicated, Ubc6 liposomes were fused to liposomes lacking Doa10 (– Doa10), or fusion was inhibited with Syb$_{sol}$ (Inhibited). After incubation with the indicated Get3 variants or buffer, liposomes were immobilized (*Figure 1—figure supplement 2B*). Input and supernatant samples were analyzed by SDS-PAGE and fluorescence scanning. Final concentrations (f.c.): 0.1 µM Ubc6, 40 nM Doa10, 10 µM Get3. (**D**) Quantification (mean ± SD) of three independent experiments as in (**C**). (**E**) Retrotranslocation of Ubc6, measured as quenching of a C-terminal AlexaFluor488 (A488) label by an antibody. Liposomes were generated as shown in (**A**). Where indicated, liposomes lacked Doa10 (– Doa10), or co-reconstitution was inhibited by using SybΔ84 (docked). Arrows indicate addition of the quenching antibody or of solubilizing amounts of detergent (Triton X-100). F.c.: 0.2 µM Ubc6, 80 nM Doa10. (**F**) Quantification (mean ± SD) of four experiments as in (**E**). The fraction of accessible dye after 30 min was compared between conditions with and without Doa10. F, normalized fluorescence. (**G**) Retrotranslocation of Ubc6, measured by a protease protection assay. Ubc6 with an N-terminal SUMO tag (SUMO-Ubc6) and a TEV protease cleavage site between the C-terminus and the fluorescent dye was used. Arrow heads indicate cleavage sites for Ulp1 and TEV protease. SUMO-Ubc6 liposomes with or without Doa10 were incubated with Ulp1. Ulp1-treated liposomes were then incubated with buffer or TEV protease. Indicated reactions contained detergent to solubilize liposomes (det). Aliquots were taken at the indicated times and analyzed by SDS-PAGE and fluorescence scanning. F.c. during incubation with TEV protease: 0.1 µM Ubc6, 40 nM Doa10, 10 µM TEV protease. (**H**) Quantification (mean ± SD) of the fraction of Ubc6 and SUMO-Ubc6 inaccessible to TEV protease, from three experiments as in (**G**). Band intensities from samples treated with TEV protease were normalized to the corresponding band intensities of samples without TEV protease.

The online version of this article includes the following source data and figure supplement(s) for figure 1:

**Source data 1.** This file contains the quantification of fluorescently labeled Ubc6 (*Figure 1D*) as well as of Rhodamine-labeled lipids (*Figure 1—figure supplement 2B*).

**Source data 2.** This file contains the quantification of the quenched fraction of Ubc6 in samples containing Doa10 compared to samples lacking Doa10, as shown in *Figure 1F*.

**Source data 3.** This file contains the quantification of the TEV-protected fraction of SUMO-Ubc6 and Ubc6 shown in *Figure 1H*.

**Figure supplement 1.** Quality control of liposomes.

**Figure supplement 1—source data 1.** This file contains the quantification of the TEV- and Ulp1 cleaved fraction of SUMO-Ubc6, as well as of Ulp1-cleaved Ubc6 that is accessible to TEV protease, as shown in *Figure 1—figure supplement 1K*.

**Figure supplement 2.** Retrotranslocation in the presence of Get3.

**Figure supplement 3.** Co-reconstitution with ATP synthase.

dye at the C-terminus. An anti-A488 antibody quenches A488 fluorescence and reports on accessibility of the C-terminus (*Figure 1E*). In the absence of Doa10, we observed a sudden decrease in fluorescence by 50% upon antibody addition, corresponding to the fraction of wrong-side out protein that exposes its C-terminus on the outside of liposomes. Upon solubilization of liposomes, the antibody quenches the fluorescence of all A488 epitopes. However, in the presence of Doa10, the sudden decrease in fluorescence was followed by a slower decrease to about 10% of the original fluorescence signal within 30 min. Thus, in the presence of Doa10, the luminally-encapsulated part of Ubc6 becomes accessible to the antibody over time. Ubc6 and Doa10 need to reside in the same membrane as we observed only minor quenching above background when we used a mutant SNARE that only supports liposome docking (*Figure 1E,F*).

As an alternative read-out for retrotranslocation, we used a protease protection assay. To identify correctly oriented Ubc6 we used an N-terminal SUMO fusion (SUMO-Ubc6) and Ulp1 protease. To monitor retrotranslocation, we introduced a TEV protease cleavage site between the C-terminus of Ubc6 and the fluorescent dye. This cleavage site resides in the liposome lumen and would only become accessible upon retrotranslocation (*Figure 1G*). Ulp1 incubation resulted in a shift of correctly oriented protein in SDS-PAGE. We then added TEV protease and followed cleavage over time (*Figure 1G,H*). Wrong-side out SUMO-Ubc6 was completely accessible to TEV protease and was cleaved independently of the presence of Doa10 within 5 min. Strikingly, only in the presence of Doa10, Ulp1-cleaved, and thus right-side out Ubc6 was also accessible to TEV cleavage over longer incubation times, indicating retrotranslocation. In liposomes lacking Doa10, only a small fraction of Ulp1-cleaved Ubc6 was accessible to TEV protease, corresponding to the not properly reconstituted Ubc6. Together, Get3 capture, antibody accessibility, and protease protection assays show that Doa10 facilitates movement of the Ubc6 TM across the membrane into the aqueous phase. Comparison of the fraction of correctly oriented protein and the released fraction shows that retrotranslocation is very efficient in all three assays. Thus, Doa10 is a retrotranslocase.

Importantly, we also tested if another unrelated multipass membrane protein leads to Ubc6 retrotranslocation by destabilizing the lipid bilayer. To this end, we purified the TF$_o$F$_1$ ATP synthase from *Bacillus* PS3, which contains 20 TM segments (*Guo et al., 2019*). Using the same reconstitution

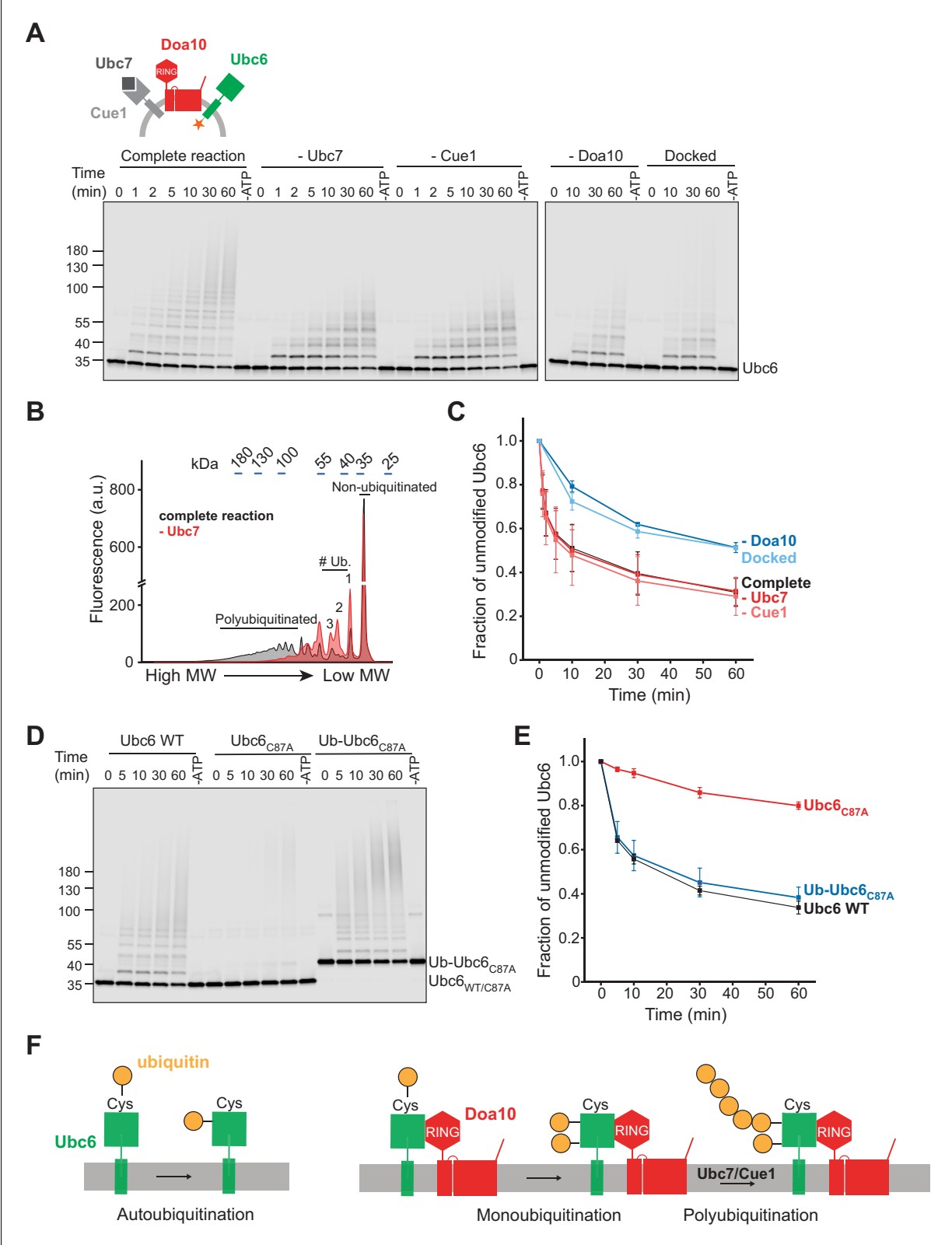

**Figure 2.** Ubiquitination of Ubc6. (**A**) Time course of ubiquitination of Ubc6. Final concentrations in the complete reaction: 40 nM Doa10, 10 nM Cue1, 1 µM Ubc7, 100 nM Ubc6, 100 nM E1, 120 µM ubiquitin, and 2.5 mM ATP. Where indicated, individual components were omitted or co-reconstitution was inhibited by using Syb∆84 (Docked). For each reaction, a 60 min sample in the absence of ATP is shown. Samples were analyzed by SDS-PAGE and fluorescence scanning. (**B**) Analysis of ubiquitin-chain length on Ubc6 from an experiment as in (**A**). Line-scans were performed on fluorescence images

*Figure 2 continued on next page*

*Figure 2 continued*

for the complete reaction and in the absence of Ubc7 at t = 30 min. Approximate molecular weights are indicated on top. # ub. denotes number of ubiquitin moieties attached. (C) Quantification (mean ± SD) of the fraction of unmodified Ubc6 from three experiments as in (A). (D) Time course of ubiquitination of Ub-Ubc6$_{C87A}$ compared to Ubc6 WT and Ubc6$_{C87A}$ in the presence of Doa10, Cue1, and Ubc7. Concentrations and analysis as in (A). (E) Quantification (mean ± SD) of the fraction of unmodified Ubc6 variants from three experiments as in (D). (F) Model for ubiquitination of Ubc6. Ubc6 autoubiquitination activity results in transfer of ubiquitin from its active site cysteine to a non-cysteine residue (*Weber et al., 2016*). In the presence of Doa10, this activity is enhanced and Ubc6 is multi-monoubiquitinated. Ubc7/Cue1 are then required to form polyubiquitin chains on monoubiquitinated Ubc6.

The online version of this article includes the following source data and figure supplement(s) for figure 2:

**Source data 1.** This file contains the quantification of the fraction of unmodified Ubc6 from three experiments as in *Figure 2A*, as shown in in *Figure 2C*.
**Source data 2.** This file contains the quantification of the fraction of unmodified Ubc6 from three experiments as in *Figure 2D*, as shown in *Figure 2E*.
**Source data 3.** This file contains the quantification of the fraction of unmodified Ubc6 from three experiments as in *Figure 2—figure supplement 1C*, as shown in *Figure 2—figure supplement 1D*.
**Figure supplement 1.** E3—independent and -dependent ubiquitination of Ubc6.

protocol as for Doa10, we co-reconstituted ATP synthase with Ubc6 (*Figure 1—figure supplement 3A,B*) and then tested for retrotranslocation of Ubc6 using the antibody accessibility assay. Under these conditions, only a minor fraction of Ubc6 becomes accessible to the antibody (*Figure 1—figure supplement 3C,D*). We conclude that retrotranslocation of Ubc6 is not due to some non-specific perturbation of the membrane caused by any multipass TM protein.

The observation that traps such as Get3 or the antibody are sufficient to drive retrotranslocation to completion suggests that Doa10 allows membrane-inserted and retrotranslocated soluble states of Ubc6 to exist in an equilibrium (*Figure 1B*). In the absence of Get3 or the anti-A488 antibody, the membrane-embedded state of the substrate is energetically favored. Get3 or the anti-A488 antibody shift the equilibrium towards the soluble state by binding to retrotranslocated Ubc6. Thus, traps bind retrotranslocated Ubc6 and prevent reinsertion. In the cell, retrotranslocation requires Cdc48 activity (*Garza et al., 2009*; *Nakatsukasa et al., 2008*; *Ye et al., 2001*), suggesting that the pulling force generated by Cdc48 provides the directionality.

To test this directly, we next investigated membrane extraction of Ubc6 by Cdc48. As Cdc48 acts on polyubiquitin chains, we first reconstituted Ubc6 polyubiquitination. Degradation of Ubc6 requires its own E2 activity, the E2 Ubc7 and its adapter, the membrane-anchored Cue1 (*Biederer et al., 1997*; *Swanson et al., 2001*; *Walter et al., 2001*). Using the fusion system, we co-reconstituted Doa10 and Cue1 with Ubc6. When we added ubiquitin activating enzyme (E1), ubiquitin, ATP, and Ubc7, we observed robust polyubiquitination of Ubc6 (complete reaction, *Figure 2A*). In the absence of Ubc7 or Cue1, polyubiquitination was abolished, and we observed ubiquitin adducts of lower molecular weight (*Figure 2A,B*). These represent multiple monoubiquitinations, as the ubiquitination pattern was very similar when we used a ubiquitin mutant in which all lysines are mutated to arginine (Ubiquitin K0) and that thus cannot form ubiquitin chains (*Figure 2—figure supplement 1A*). Monoubiquitination also occurred in the absence of Doa10 (*Figure 2—figure supplement 1B*), but was enhanced in its presence (*Figure 2—figure supplement 1C,D*). Kinetics of transfer of the first ubiquitin onto Ubc6 were independent of the presence of Ubc7/Cue1 (*Figure 2C*), suggesting that monoubiquitination is a prerequisite for Ubc7-dependent polyubiquitination. This is indeed the case, as a catalytically inactive Ubc6 mutant (Ubc6$_{C87A}$) was not ubiquitinated, but an N-terminal fusion of ubiquitin with inactive Ubc6 (Ub-Ubc6$_{C87A}$) was a substrate for Ubc7-dependent polyubiquitination (*Figure 2D,E* and *Figure 2—figure supplement 1E*). Together, these results establish that after active site loading of Ubc6 with ubiquitin, Doa10 catalyzes Ubc6-monoubiquitination, followed by Ubc7/Cue1-dependent polyubiquitination (*Figure 2F*). These results agree with observations made in intact cells and with recombinant soluble fragments of Doa10 and Ubc6 (*Walter et al., 2001*; *Weber et al., 2016*). Furthermore, they indicate that our reconstituted system faithfully recapitulates the in vivo ubiquitination pathway for Ubc6.

Next, we tested for membrane extraction of polyubiquitinated Ubc6 by the Cdc48 ATPase. To this end, we immobilized Doa10/Ubc6 liposomes after the ubiquitination reaction, then incubated with Cdc48 and its co-factors Ufd1 and Npl4 (UN), and analyzed soluble and membrane-bound fractions (*Figure 3A* and *Figure 3—figure supplement 1,A* to D). We observed Cdc48- and ubiquitination-dependent extraction of Ubc6 (*Figure 3A,B* and *Figure 3—figure supplement 1E,F*).

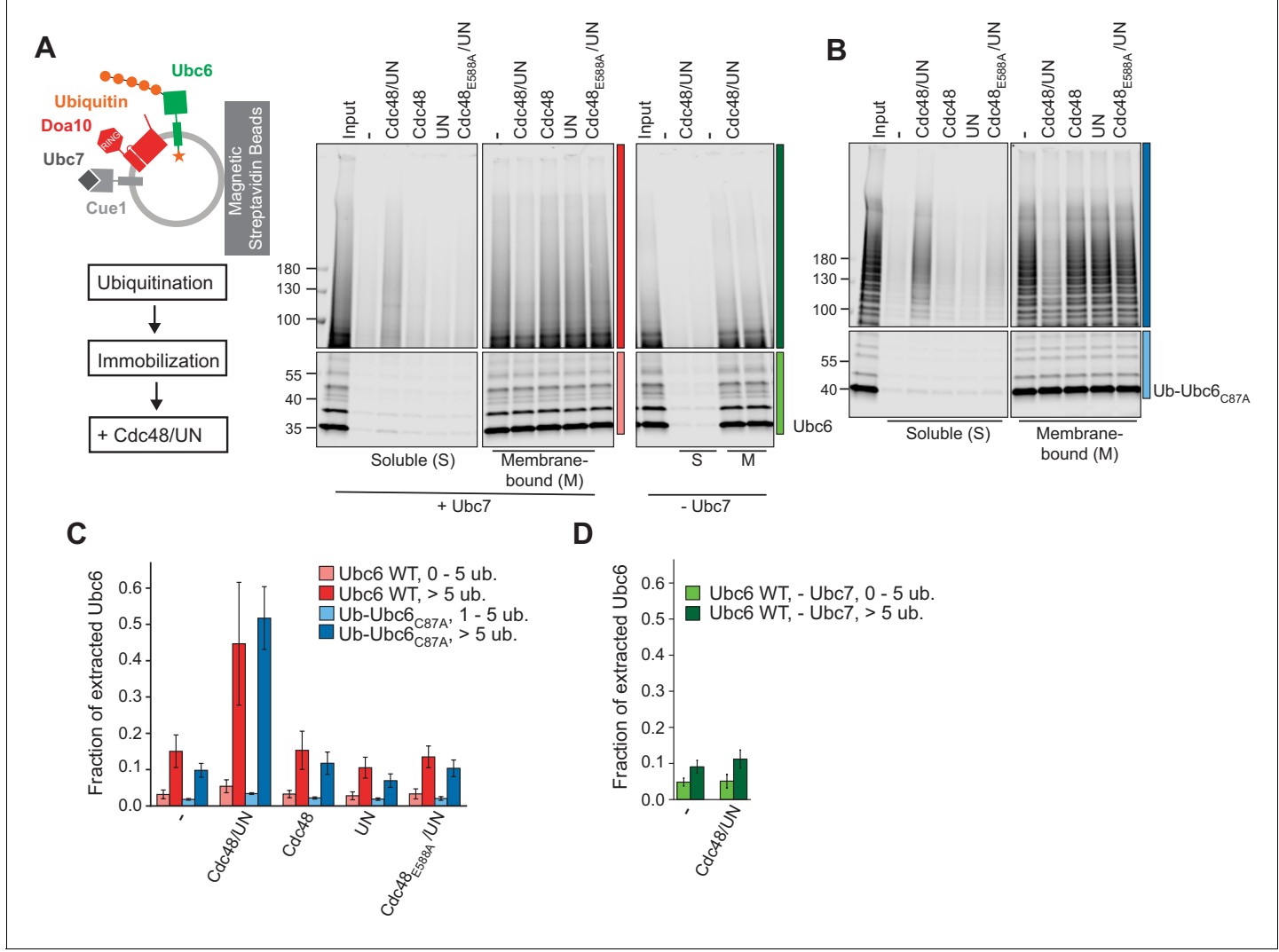

**Figure 3.** Cdc48-mediated Membrane Extraction of Ubc6. (**A**) Extraction of Ubc6 by Cdc48 and Ufd1/Npl4 (UN). After ubiquitination, liposomes were immobilized (*Figure 3—figure supplement 1,A* to D). One bead equivalent was removed, and bound protein was eluted with SDS sample buffer (Input). Beads were then incubated with the indicated components. Soluble (S) and membrane-bound (M) material were analyzed by SDS-PAGE and fluorescence scanning. Colored bars indicate categorization of ubiquitin chain length as used for quantification in (**C**) and (**D**). For better visibility, bottom and top gel parts are scaled differently. See *Figure 3—figure supplement 1E* for uncut image. Final concentrations: 50 nM Ubc6, 20 nM Doa10, 0.1 μM Cdc48 hexamer, 0.1 μM Ufd1 and Npl4. (**B**) As in (**A**), but with Ub-Ubc6$_{C87A}$ instead of Ubc6. See *Figure 3—figure supplement 1F* for uncut image. (**C**) Quantification (mean ± SD) of three experiments as in (**A**) and (**B**). Ubiquitinated species were categorized according to ubiquitin chain length, as indicated in (**A**) and (**B**). The signal in the soluble fraction was normalized to that in the input. (**D**) Quantification (mean ± SD) of three experiments as in (**A**), when ubiquitination was performed in the absence of Ubc7.

The online version of this article includes the following source data and figure supplement(s) for figure 3:

**Source data 1.** This file contains the quantification of the fraction of extracted Ubc6 from three experiments as in *Figure 3A,B*, as shown *Figure 3C,D*.

**Figure supplement 1.** Cdc48-mediated extraction.

Extraction efficiency was dependent on the length of ubiquitin chains, with five ubiquitin moieties being minimally required (*Figure 3—figure supplement 1G*). In the presence of Cdc48/UN, 45 ± 17% of Ubc6 molecules with more than five attached ubiquitin moieties were extracted, compared to 15 ± 4% in the absence of the Cdc48 complex (*Figure 3A,C*). No extraction above this background was observed when either Cdc48 or Ufd1/Npl4 were omitted. Furthermore, ATP hydrolysis by the Cdc48 complex was necessary, as Cdc48$_{E588A}$ was inactive. Polyubiquitin chains were required because we found no extraction above background when ubiquitination was performed in the absence of Ubc7 or less than five ubiquitins were attached (*Figure 3A,C,D*). Similar observations

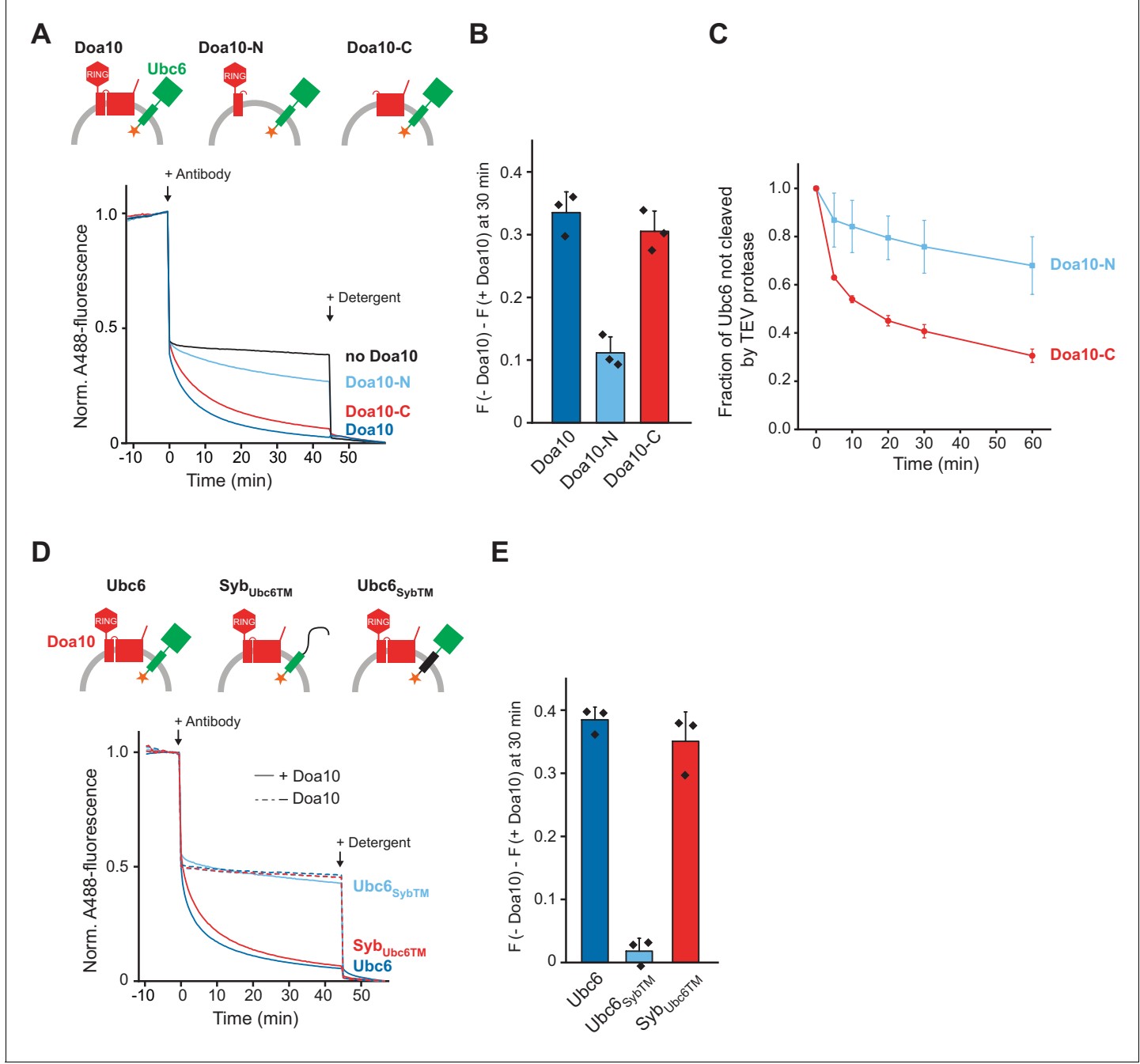

**Figure 4.** Structural Determinants for Retrotranslocation. (**A**) Retrotranslocation of Ubc6 by Doa10 variants, as measured by accessibility of a fluorescence quenching antibody to a C-terminal A488 dye on Ubc6, as described in ***Figure 1E***. Ubc6 liposomes containing the indicated Doa10 variants were used. Doa10-N, residues 1–468; Doa10-C, residues 434–1319. Arrows indicate addition of antibody or detergent. Final concentrations (f. c.): 0.2 µM Ubc6, 80 nM Doa10 variants. (**B**) Quantification (mean ± SD) of three experiments as in (**A**). The fraction of accessible dye after 30 min was compared between conditions with the indicated Doa10 variant and without Doa10. F, normalized fluorescence. (**C**) Retrotranslocation of Ubc6 by Doa10 variants, as measured by accessibility of TEV protease to the C-terminus of Ubc6, as described in ***Figure 1G***. SUMO-Ubc6 liposomes with either Doa10-N or Doa10-C were treated with Ulp1 to identify right-side out oriented Ubc6. TEV protease was added and samples at different time points were analyzed by SDS-PAGE and fluorescence scanning. Quantification as in ***Figure 1H***, but only for Ulp1-cleaved Ubc6. F.c. during incubation with TEV protease: 0.1 µM Ubc6, 40 nM Doa10 variants, 10 µM TEV protease. (**D**) Retrotranslocation of Ubc6 variants measured as in (**A**). A488-labeled Ubc6, Ubc6$_{SybTM}$, or Syb$_{Ubc6TM}$ were directly co-reconstituted with Doa10 because Syb$_{Ubc6TM}$ was incompatible with SNARE-mediated co-reconstitution. Liposomes containing Doa10 were affinity-purified for this experiment (***Figure 4—figure supplement 1A,B***). (**E**) Quantification (mean ± SD) of three experiments as in (**D**). The fraction of accessible dye after 30 min was compared between conditions with and without Doa10.

*Figure 4 continued on next page*

*Figure 4 continued*

The online version of this article includes the following source data and figure supplement(s) for figure 4:

**Source data 1.** This file contains the quantification of the quenched fraction of Ubc6 in samples containing Doa10 or its variants compared to samples lacking Doa10 from three experiments as in *Figure 4A*, as shown in *Figure 4B*.

**Source data 2.** This file contains the quantification of the TEV-protected fraction of Ubc6 from three experiments, as shown in *Figure 4C*.

**Source data 3.** This file contains the quantification of the quenched fraction of Ubc6 or its variants in samples containing Doa10 compared to samples lacking Doa10 from three experiments as in *Figure 4D*, as shown in *Figure 4E*.

**Figure supplement 1.** Antibody accessibility assay for Ubc6/Syb chimera.

**Figure supplement 1—source data 1.** This file contains numerical values for data shown in *Figure 4—figure supplement 1E*.

---

were made when we used Ub-Ubc6$_{C87A}$ instead of WT Ubc6 to increase the efficiency of polyubiquitination (*Figure 3B,C*). Together, these observations show that the Cdc48 complex provides the driving force for the extraction of a polyubiquitinated membrane protein. We currently do not understand what limits the efficiency of Cdc48 mediated extraction in our assay. It is possible that a stabilizing chaperone or an accessory factor such as the Cdc48 co-factor Ubx2 would contribute to complete extraction (*Neuber et al., 2005*; *Schuberth and Buchberger, 2005*).

To define structural elements in Doa10 important for its retrotranslocase activity, we generated two truncated versions of Doa10 that encompassed either only the N-terminal RING domain and the first two TM segments (Doa10-N), or the C-terminal part containing TM segments 3–14 (Doa10-C) (*Figure 1—figure supplement 1,F* to H). The sites of truncation were chosen based on the finding that in the yeast *Kluyveromyces lactis*, Doa10 is expressed as two separate polypeptides with similar boundaries (*Stuerner et al., 2012*). We then tested if those Doa10 variants retrotranslocate Ubc6 using the antibody accessibility assay. Doa10-C behaved similarly to full-length Doa10, whereas Doa10-N resulted in only minor quenching above background (*Figure 4A,B*). Corresponding observations were made when we tested for retrotranslocation using the protease protection assay (*Figure 4C*). These results show that TM segments 3–14 in Doa10 are sufficient to mediate retrotranslocation of Ubc6.

To test for structural elements in Ubc6 relevant for retrotranslocation, we generated mutants in which either its TM anchor or its cytoplasmic part were replaced with the corresponding segments of the TA protein synaptobrevin (Ubc6$_{SybTM}$ and Syb$_{Ubc6TM}$, respectively). We then tested for retrotranslocation of these mutants using the antibody accessibility assay (*Figure 4—figure supplement 1A,B*). We only observed retrotranslocation of the Ubc6 TM, but not of the Syb TM (*Figure 4D,E*). A similar experimental setup also allowed us to exclude leakage or liposome rupture as the cause for antibody accessibility (*Figure 4—figure supplement 1,C* to E). Thus, the identity of the substrate's TM segment is important for retrotranslocation.

Next, we tested how these mutations in the TM domains of Doa10 and Ubc6 affect ubiquitination of Ubc6. To specifically test for effects on polyubiquitination, we again used Ub-Ubc6$_{C87A}$, for which the initial ubiquitination steps (ubiquitin loading and monoubiquitination) are bypassed. Replacement of the Ubc6 TM with the TM of Syb mildly affected polyubiquitination as seen by the emergence of shorter ubiquitin chains in the case of the Syb TM (*Figure 5A,B*; *Figure 5—figure supplement 1A,B*). To test for effects of TM replacement on monoubiquitination of Ubc6, we compared Ubc6 and Ubc6$_{SybTM}$. Doa10-dependent monoubiquitination of Ubc6$_{SybTM}$ was impaired (*Figure 5C,D*; *Figure 5—figure supplement 1C,D*), while E3-independent autoubiquitination of this mutant was unaffected (*Figure 5—figure supplement 1E,F*). Thus, the Ubc6 TM anchor contributes to the efficient Doa10-dependent ubiquitination of Ubc6, indicating a more efficient recruitment to Doa10.

Moreover, efficient ubiquitination of Ubc6 requires the TM domain of Doa10. Ubc6 polyubiquitination by Doa10-N was less efficient compared to full-length Doa10 (*Figure 5E,F*). This was not due to impaired E3 activity, because co-reconstitution of Doa10-N and Doa10-C together restored ubiquitination to WT levels. Monoubiquitination in the absence of Ubc7 was similarly affected (*Figure 5—figure supplement 2,A* to C). We conclude that the Doa10 region that includes TMs 3–14 plays a role in ubiquitination of Ubc6. Both ubiquitination and retrotranslocation of Ubc6 are sensitive to changes in the membrane-embedded regions of Ubc6 and Doa10, indicating a specific interaction of Doa10 with the TM domain of Ubc6. Previous observations suggested that the identity of the TM

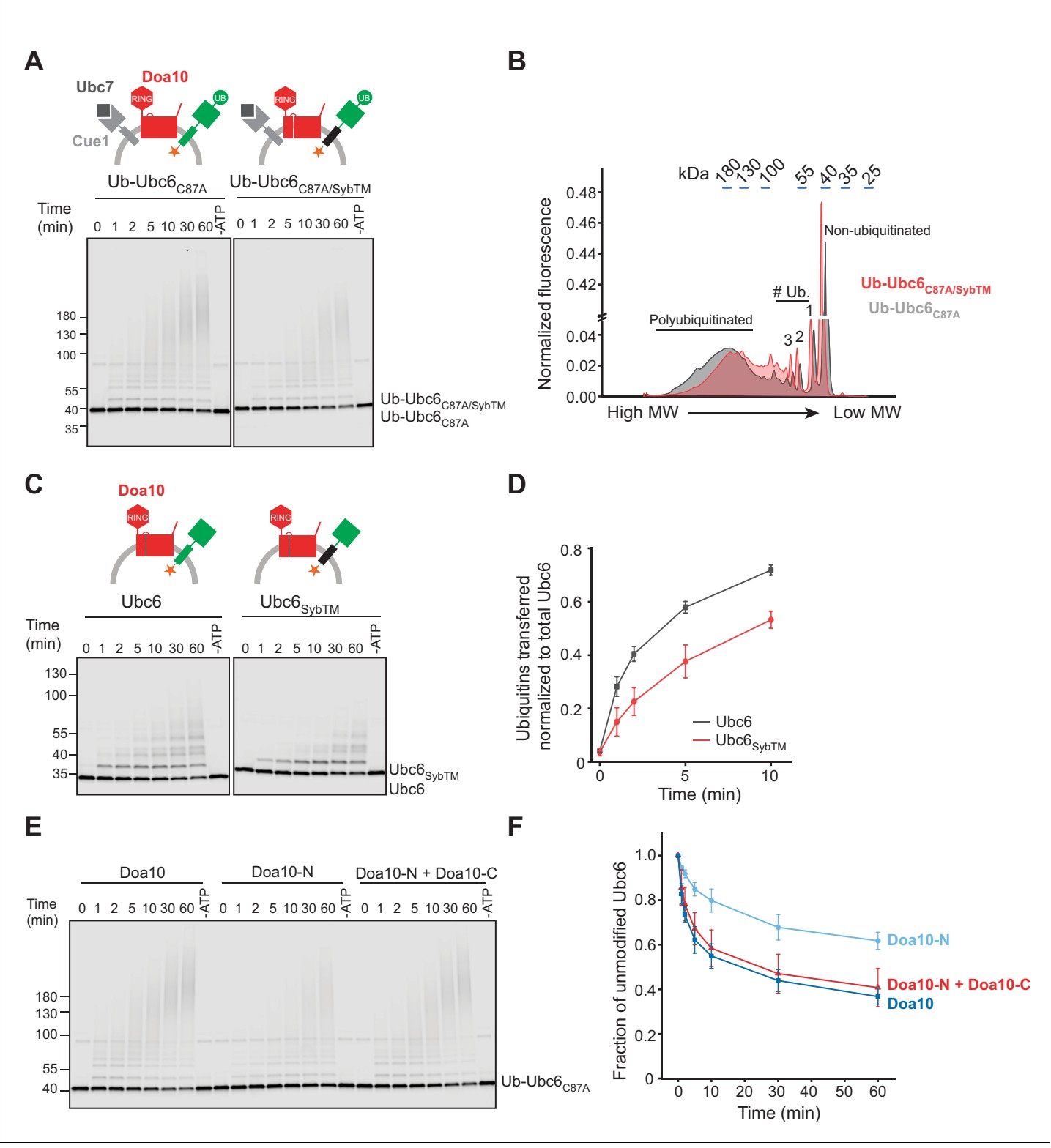

**Figure 5.** Structural Determinants for Ubiquitination. (**A**) Time course of ubiquitination of Ub-Ubc6$_{C87A}$ or Ub-Ubc6$_{C87A/SybTM}$ by Doa10 in the presence of Cue1/Ubc7. For each reaction, a 60 min sample in the absence of ATP is shown. Samples were analyzed by SDS-PAGE and fluorescence scanning. Final concentrations: 40 nM Doa10, 10 nM Cue1, 1 μM Ubc7, 100 nM Ubc6 variants, 100 nM E1, 120 μM ubiquitin, and 2.5 mM ATP. See *Figure 5—figure supplement 1A* for quantification of unmodified Ubc6 variants. (**B**) Comparison of ubiquitin-chain length on Ub-Ubc6$_{C87A}$ or Ub-Ubc6$_{C87A/SybTM}$. Line-scans were performed on fluorescence images of two representative gel samples (30 min timepoint) as in (**A**). Approximate molecular weights are

*Figure 5 continued on next page*

*Figure 5 continued*

indicated on top. # Ub., number of ubiquitin moieties attached. (C) Time-course of Ubc6 WT or Ubc6$_{SybTM}$ ubiquitination in the absence of Ubc7/Cue1. Analysis and concentrations as in (A). See *Figure 5—figure supplement 1C* for quantification of unmodified Ubc6 variants. (D) Quantification (mean ± SD) of total ubiquitin-transfer to Ubc6 or Ubc6$_{SybTM}$ from three experiments as in (C). Intensities of Ubc6 variants with one to four ubiquitin moieties attached were determined as described in *Figure 5—figure supplement 1D*, summed up for each time point and normalized to total Ubc6 in the reaction. (E) Time course of ubiquitination of Ub-Ubc6$_{C87A}$ by Doa10 variants in the presence of Cue1/Ubc7. Liposomes contained Ub-Ubc6$_{C87A}$ and either full-length Doa10, only Doa10-N, or both Doa10-N and -C. Analysis and concentrations as in (A). (F) Quantification (mean ± SD) of unmodified Ub-Ubc6$_{C87A}$ from three experiments as in (E).

The online version of this article includes the following source data and figure supplement(s) for figure 5:

**Source data 1.** This file contains the quantification of the number of ubiquitins (n) transferred per Ubc6 or Ubc6$_{SybTM}$ (*Figure 5—figure supplement 1D*) as well as the quantification of the number of total ubiquitin transferred from three experiments as in *Figure 5C*, as shown in *Figure 5D*.

**Source data 2.** This file contains the quantification of the fraction of unmodified Ubc6 from three experiments as in *Figure 5E*, as shown in *Figure 5F*.

**Figure supplement 1.** Ubiquitination of Ubc6/Syb chimera.

**Figure supplement 1—source data 1.** This file contains the quantification of the fraction of unmodified Ubc6 as shown in *Figure 5—figure supplement 1A, C and F*.

**Figure supplement 2.** Ubc6 ubiquitination by Doa10 variants.

**Figure supplement 2—source data 1.** This file contains the quantification of the fraction of unmodified Ubc6 as shown in *Figure 5—figure supplement 2B*.

**Figure supplement 2—source data 2.** This file contains the quantification of the number of total ubiquitin transferred in presence of different Doa10 variants from three experiments, as shown in *Figure 5—figure supplement 2C*.

segment played a role in substrate degradation (*Habeck et al., 2015*; *Ruggiano et al., 2016*; *Walter et al., 2001*). Our results show that the TM domain of Doa10 recognizes substrates and thereby contributes to the specificity of substrate selection. The observation that the mutant version of Ubc6 (Ubc6$_{SybTM}$) is still ubiquitinated to some extent suggests that other factors might contribute to substrate discrimination. Deubiquitinating enzymes have previously been shown to sharpen substrate selectivity in ERAD and might also play such a role in the context of Doa10-mediated ERAD (*Zhang et al., 2013*).

Substrates of Doa10 exhibit a wide range of topologies. They may contain multiple TM segments, such as the misfolded variants of the multi-spanning membrane proteins Pma1 and Ste6, called Pma* and Ste6*, respectively (*Huyer et al., 2004*; *Wang and Chang, 2003*), or luminal folded domains (*Vashist and Ng, 2004*). We next asked the question how the presence of an additional luminal polypeptide segment or an interaction with another luminal protein affects retrotranslocation. We appended a streptavidin binding peptide (SBP) to the C-terminus of Ubc6 (Ubc6-SBP), and formed a complex with streptavidin (*Figure 6—figure supplement 1A,B*). When we co-reconstituted this complex with Doa10, we observed no quenching over time upon antibody addition (*Figure 6A, B*). When we added biotin, which breaks the high affinity SBP-streptavidin interaction (*Keefe et al., 2001*), we observed Doa10-dependent quenching over time. This was only the case when biotin was used, but much reduced when we used a biotinylated protein, which is still capable of dissociating streptavidin from Ubc6-SBP on the outside of liposomes (*Figure 6—figure supplement 1, C to E*), but cannot pass the membrane. Together, this shows that a protein-protein interaction on the luminal side of the membrane, mimicking the presence of a folded domain, acts as an anchor and prevents retrotranslocation of Ubc6.

Finally, we tested if this anchoring can be overcome by the Cdc48 complex. Liposomes containing Doa10 and Ubc6-SBP in complex with streptavidin were incubated with ubiquitination mix followed by the addition of Cdc48 complex and anti-A488 antibody. Retrotranslocation occurred depending on polyubiquitination and Cdc48 activity (*Figure 6C,D* and *Figure 6—figure supplement 1F*). In the absence of ubiquitin, when Ubc7 was omitted, or when we used the catalytically inactive Cdc48$_{E588A}$, no fluorescence quenching above background was observed. Importantly, streptavidin remained encapsulated in liposomes in reactions where Ubc6-SBP was extracted (*Figure 6E,F*). Thus, Cdc48 action on the cytosolic side of the membrane leads to dissociation of streptavidin from the SBP-tag in the liposome lumen. As this reaction entails the breaking of bonds that are comparable to the intramolecular interactions that keep a protein folded, we interpret the dissociation of the SBP tag from streptavidin as unfolding. Our results thus show that Doa10 retrotranslocates a luminal protein segment in an unfolded state. Cdc48, acting on cytoplasmic polyubiquitin chains, generates a

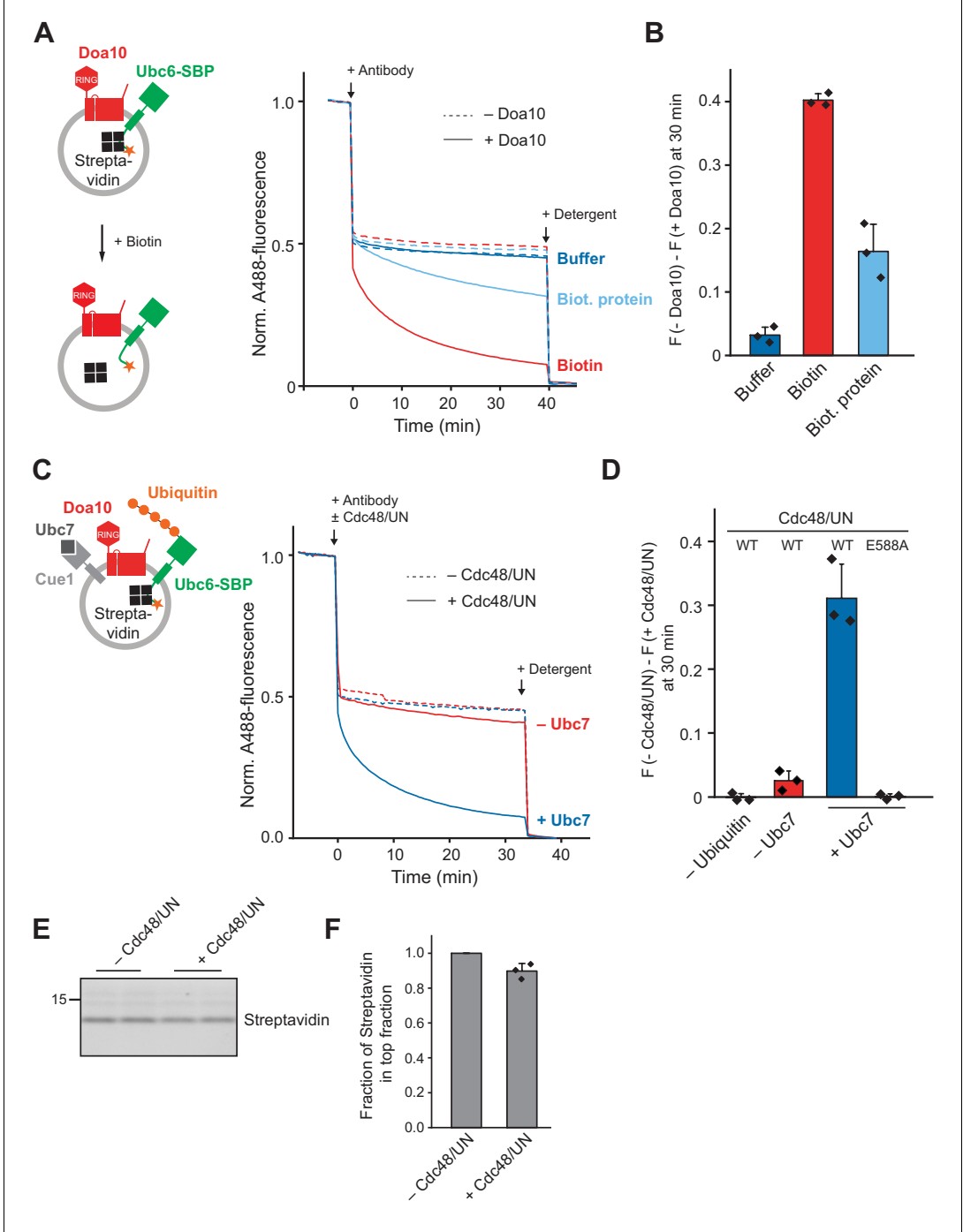

**Figure 6.** Effect of a Luminal Domain on Retrotranslocation. (**A**) Retrotranslocation of Ubc6 with a C-terminal streptavidin binding peptide (SBP), reconstituted in complex with streptavidin, was measured by accessibility of a fluorescence quenching antibody to a C-terminal A488 dye on Ubc6, as described *Figure 1E*. Liposomes were incubated with buffer, biotin or a biotinylated protein prior to addition of the antibody. Final concentrations: 0.2 μM Ubc6, 80 nM Doa10, 0.25 μM streptavidin, 1.5 μM biotin or biotinylated protein. (**B**) Quantification (mean ± SD) of three experiments as in (**A**). The fraction of accessible dye after 30 min was compared between conditions with and without Doa10. F, normalized fluorescence. (**C**) Effect of Cdc48 and Ufd1/Npl4 (UN) on retrotranslocation of Ubc6-SBP in complex with streptavidin, measured using the antibody accessibility assay as in (**A**). Prior to the fluorescence measurement, liposomes were incubated with ubiquitination mix with or without Ubc7. Arrows indicate when antibody, with or without Cdc48/UN, or detergent were added. Final concentrations: 0.17 μM Ubc6-SBP, 68 nM Doa10, 0.17 μM hexameric Cdc48, Ufd1, and Npl4. See *Figure 6—figure supplement 1F* for gel samples of ubiquitination reaction. (**D**) Quantification (mean ± SD) of three experiments as in (**C**). The fraction of accessible dye after 30 min was compared between conditions with and without Cdc48/UN. In addition, experiments lacking ubiquitin or with the Cdc48 mutant E588A were quantified. (**E**) Determination of liposome-encapsulated streptavidin after extraction. Samples from experiments as in (**D**)

*Figure 6 continued on next page*

*Figure 6 continued*

were taken at t = 30 min. Biotin was added, and liposomes floated in a Nycodenz gradient. Co-floating streptavidin was detected in SDS-PAGE using stain-free technology. Two replicates are shown for each condition. (F) Quantification (mean ± SD) of the relative amount of streptavidin co-floating from three experiments as in (E). Each data point represents the mean of two replicates as shown in (E).

The online version of this article includes the following source data and figure supplement(s) for figure 6:

**Source data 1.** This file contains the quantification of the quenched fraction of Ubc6 in samples containing Doa10 compared to samples lacking Doa10 from three experiments as in *Figure 6A*, as shown in *Figure 6B*.

**Source data 2.** This file contains the quantification of the quenched fraction of Ubc6 in samples containing Cdc48/UN compared to samples lacking Cdc48/UN from three experiments as in *Figure 6C*, as shown in *Figure 6D*.

**Source data 3.** This file contains the quantification of streptavidin in the top flotation fraction from three experiments as in *Figure 6E*, as shown in *Figure 6F*.

**Figure supplement 1.** Liposomes with Ubc6-SBP and streptavidin.

---

mechanical force that results in luminal unfolding und drives retrotranslocation. The integrity of the membrane is maintained in this process.

## Discussion

During extraction of a protein from the membrane, an energetic barrier must be overcome that depends on the hydrophobicity of its TM domain (*Botelho et al., 2013*; *Guerriero et al., 2017*). Our results provide evidence that Doa10 contributes to overcoming the energetic barrier for membrane protein extraction. This is demonstrated by passive movement of a tail-anchored membrane protein into the aqueous phase in the presence of Doa10. In the absence of a folded luminal domain, factors such as Get3 or an antibody that trap the retrotranslocated state are sufficient to drive the reaction, bypassing the requirement for ubiquitination and Cdc48.

ERAD also entails retrotranslocation of less hydrophobic sequences, such as those in luminal loops or domains, across the hydrophobic core of the membrane. We show that, once dissociated from streptavidin, a 57 amino acid long luminal protein segment that includes the SBP-tag is retrotranslocated by Doa10 without markedly affecting retrotranslocation kinetics. Thus, Doa10 can also accommodate these less hydrophobic sequences in the retrotranslocation process.

A major unresolved question concerns the fate of luminal domains during retrotranslocation. Some studies suggested that luminal domains cross the ER membrane in a folded state, based either on the observation that substrates containing tightly folded domains were retrotranslocated in the first place or that they were detected in a folded state after retrotranslocation into the cytosol (*Fiebiger et al., 2002*; *Petris et al., 2014*; *Shi et al., 2019*; *Tirosh et al., 2003*). Others suggested that retrotranslocation requires unfolding of luminal protein segments, including reduction of disulfide bonds, prior to retrotranslocation (reviewed in *Ellgaard et al., 2018*). Some of these differences might be explained by the fact that different ubiquitin ligase complexes were involved that might have different requirements for retrotranslocation. Our data show that Doa10 does not accommodate a folded domain during retrotranslocation. Instead, unfolding of polypeptide segments on the luminal side of the membrane is a direct consequence of Cdc48 acting on cytoplasmically attached polyubiquitin chains.

It is unclear if chaperone-driven retrotranslocation of Ubc6 in the absence of ubiquitination or Cdc48 action also occurs in the cell. Genetic and biochemical experiments in yeast showed that degradation of TM domain containing Doa10 substrates, including Ubc6, strictly depends on Cdc48, Ufd1 and Npl4 (*Foresti et al., 2013*; *Habeck et al., 2015*; *Huyer et al., 2004*; *Neuber et al., 2005*; *Ravid et al., 2006*; *Ruggiano et al., 2016*; *Wang and Chang, 2003*). In the case of the strongly hydrophobic, multi-spanning Doa10 substrate Ste6*, fractionation experiments also showed that the Cdc48 complex is required for the retrotranslocation step (*Nakatsukasa et al., 2008*; *Nakatsukasa and Kamura, 2016*; *Neal et al., 2018*). However, these experiments do not exclude the possibility that a relatively mildly hydrophobic protein such as Ubc6 retrotranslocates into the cytosol in a Doa10-dependent, but ubiquitination- and Cdc48-independent manner. Speculatively, such a chaperone-stabilized cytoplasmic pool would not be a substrate for proteasomal degradation but rather for chaperone-assisted reinsertion into the ER membrane, and might therefore be difficult to detect. Furthermore, as chaperones would only be able to capture a substrate that has emerged

from the membrane, we expect such chaperone-driven retrotranslocation to be sensitive to the overall hydrophobicity of the TM segment. Interestingly, several studies showed that chaperones play an important role in ERAD of membrane proteins at different stages of the ERAD process. In yeast, Hsp70 and Hsp40 chaperones promote ubiquitination of Ste6* and a misfolded variant of Pma1 (*Han et al., 2007*; *Nakatsukasa et al., 2008*). In mammals, the Bag6-Ubl4a-Trc35 chaperone complex facilitates ERAD of membrane proteins by stabilizing a soluble cytoplasmic state (*Claessen and Ploegh, 2011*; *Claessen et al., 2014*; *Ernst et al., 2011*; *Wang et al., 2011*; *Xu et al., 2012*). If such chaperones also act as a sink in a putative Cdc48-independent retrotranslocation process, remains to be determined.

Using a reconstituted system, we disentangled different ERAD subreactions, that is substrate recruitment, ubiquitination, retrotranslocation, and membrane extraction. This allowed us to identify two activities of Doa10: retrotranslocase and ubiquitin ligase. The interplay of Doa10 and the Cdc48 complex during the extraction process remains to be further explored. Studies showed that Cdc48 is recruited to the Doa10-complex via Ubx2 (*Neuber et al., 2005*; *Schuberth and Buchberger, 2005*). This recruitment is dependent on the ubiquitination activity of Doa10, suggesting that Cdc48-recruitment requires either autoubiquitination of Doa10 or substrate ubiquitination. Moreover, the Derlin Dfm1, which also interacts with Cdc48 through its carboxy-terminal SHP-box, has been shown to be required for degradation of Ste6* (*Neal et al., 2018*). Our experimental system is expandable and can be used to explore how Dfm1, Ubx2 or other factors affect retrotranslocation by Doa10 and Cdc48.

How exactly Doa10 facilitates release of proteins from the membrane is unclear. Structural information on Doa10 is necessary to further elucidate its mechanism of action. We speculate that TM segments access a binding site in Doa10 through a lateral gate. This might promote delipidation of TM segments and breaking of helix-helix interactions in multi-spanning membrane proteins. Quality control pathways for membrane proteins that require their extraction from the membrane exist not only in the ER, but also in other organelles. In the Golgi, mitochondria and chloroplasts, membrane proteins are removed from the organelle for proteasomal degradation in the cytosol. These processes are dependent on ubiquitination and Cdc48 (*Heo et al., 2010*; *Ling et al., 2019*; *Schmidt et al., 2019*; *Tanaka et al., 2010*). Moreover, extraction of membrane proteins also occurs by membrane-bound AAA ATPases that often have not only unfolding, but also proteolytic activity. In mitochondria, Msp1 and the FtsH-related AAA metalloproteases m-AAA and i-AAA are examples for membrane-bound AAA ATPases (*Glynn, 2017*). For m-AAA mediated extraction into the mitochondrial matrix, a contribution of the TM domain was shown, suggesting a retrotranslocase activity (*Korbel et al., 2004*; *Lee et al., 2017*). We propose that membrane-bound retrotranslocases generally contribute to AAA protein-driven extraction of membrane proteins.

## Materials and methods

### Key resources table

| Reagent type (species) or resource | Designation | Source or reference | Identifiers | Additional information |
|---|---|---|---|---|
| Gene (*S. cerevisiae*) | DOA10 | | YIL030C | Amplified from BY4741 |
| Gene (*S. cerevisiae*) | UBC6 | | YER100W | Amplified from BY4741 |
| Gene (*S. cerevisiae*) | UBA1 | | YKL210W | Amplified from BY4741 |
| Gene (*S. cerevisiae*) | UBC7 | | YMR022W | Amplified from BY4741 |
| Gene (*S. cerevisiae*) | CUE1 | | YMR264W | Amplified from BY4741 |
| Gene (*S. cerevisiae*) | CDC48 | | YDL126C | Amplified from BY4741 |
| Gene (*S. cerevisiae*) | UFD1 | | YGR048W | Amplified from BY4741 |
| Gene (*S. cerevisiae*) | NPL4 | | YBR170C | Amplified from BY4741 |
| Gene (*S. cerevisiae*) | GET3 | | YDL100C | Amplified from BY4741 |
| Gene (*Rattus norvegicus*) | Synaptobrevin 2 | | NP_036795 | |

*Continued on next page*

*Continued*

| Reagent type (species) or resource | Designation | Source or reference | Identifiers | Additional information |
|---|---|---|---|---|
| Strain, strain background (*S. cerevisiae*) | BY4741 | GE Dharmacon | | MATa *his3Δ1 leu2Δ0 met15Δ0 ura3Δ0* |
| Strain, strain background (*S. cerevisiae*) | Δdoa10 | GE Dharmacon | | MATa *his3Δ1 leu2Δ0 met15Δ0 ura3Δ0 doa10::kanR* |
| Strain, strain background (*E. coli*) | BL21 (DE3) | New England Biolabs | C2527I | Competent Cells |
| Strain, strain background (*E. coli*) | BL21-CodonPlus (DE3)-RIPL | Agilent | # 230280 | Competent Cells |
| Antibody | Anti-SBP (clone 20), mouse monoclonal | Merck | Cat#: MAB10764 | (1:2500) diluted in 5% milk TBS-T |
| Antibody | Anti-His$_6$ (Clone13/45/31-2), mouse monoclonal | Dianova | Cat#: DIA-900 | (1:500) diluted in 2% BSA PBS-T |
| Antibody | Goat polyclonal anti-mouse IgG secondary antibody (IRDye 800 CW) | Li-Cor Biosciences | Cat#: 926–32210 RRID:AB_2687825 | (1:15000) |
| Antibody | Goat polyclonal anti-mouse IgG secondary antibody (IRDye 680 RD) | Li-Cor Biosciences | Cat# 926–68070, RRID:AB_10956588 | (1:15000) |
| Antibody | Rabbit polyclonal anti-Alexa Fluor 488 | Thermo Fisher Scientific | Cat# A-11094f, RRID:AB_221544 | (1:15) diluted |
| Peptide, recombinant protein | Streptavidin | New England Biolabs | N7021S | |
| Peptide, recombinant protein | Gly-Gly-Gly-Cys peptide | Thermo Fisher Scientific | | for Sortase-mediated labeling |
| Commercial assay or kit | Gibson Assembly Master Mix | New England Biolabs | E2611S | |
| Commercial assay or kit | Q5 Site-Directed Mutagenesis Kit | New England Biolabs | E0554S | |
| Commercial assay or kit | MasterPure Yeast DNA Purification Kit | Epicentre (Lucigen) | MPY80200 | |
| Chemical compound, drug | Decyl Maltose Neopentyl Glycol (DMNG) | Anatrace | NG322 | |
| Chemical compound, drug | GDN | Anatrace | GDN101 | |
| Chemical compound, drug | n-Octyl β-D-glucopyranoside (OG) | Glycon Biochemicals | D97001 | |
| Chemical compound, drug | n-Decyl β-D-Maltopyranoside (DM) | Glycon Biochemicals | D99003 | |
| Chemical compound, drug | Dodecyl-β-D-maltoside (DDM) | Carl Roth | CN26.5 | |
| Chemical compound, drug | Anapoe-X-100 (Triton X-100) | Anatrace | APX100 | |
| Chemical compound, drug | Sodium cholate hydrate | Sigma | C1254 | |
| Chemical compound, drug | 1-palmitoyl-2-oleoyl-glycero3-phospho-choline (POPC) | Avanti Polar Lipids | 850457P | |
| Chemical compound, drug | 1,2-dioleoyl-sn-glycero-3-phosphoethanolamine (DOPE) | Avanti Polar Lipids | 850725P | |

*Continued on next page*

*Continued*

| Reagent type (species) or resource | Designation | Source or reference | Identifiers | Additional information |
|---|---|---|---|---|
| Chemical compound, drug | 1,2,-dioleoyl-sn-glycero-3-phospho-L-serine (DOPS) | Avanti Polar Lipids | 840035P | |
| Chemical compound, drug | 1,2-dioleoyl-sn-glycero-3-phosphoethanolamine-N-(biotinyl) | Avanti Polar Lipids | 870282P | |
| Chemical compound, drug | 1,2-dioleoyl-sn-glycero-3-phosphoethanolamine-N-(lissamine rhodamine B sulfonyl) (Rhd-PE) | Avanti Polar Lipids | 810150P | |
| Chemical compound, drug | Ergosterol (>95%, HPLC) | Sigma-Aldrich | 45480 | |
| Chemical compound, drug | ATP | PanReac AppliChem | A1348 | |
| Chemical compound, drug | AlexaFluor488 maleimide | Thermo Fisher Scientific | A10254 | |
| Chemical compound, drug | DyLight 680 maleimide | Thermo Fisher Scientific | 46618 | |
| Chemical compound, drug | Pierce Detergent removal spin columns | Thermo Fisher Scientific | 87777 | |
| Chemical compound, drug | Ubiquitin (WT), yeast | Boston Biochem | U-100Sc | |
| Chemical compound, drug | Ubiquitin (K0) | LifeSensors | SI209 | |
| Chemical compound, drug | YEP broth | Formedium | CCM0410 | |
| Chemical compound, drug | Yeast Nitrogen Base (YNB) | US Biological Life Sciences | C19032801 | |
| Chemical compound, drug | CSM,-Ura | Formedium | DCS0161 | |
| Chemical compound, drug | D-(+)-Galactose | PanReac AppliChem | A1131 | |
| Other | HisPur NiNTA resin | Thermo Fisher Scientific | 88223 | |
| Other | Pierce High Capacity Streptavidin Agarose | Thermo Fisher Scientific | 20361 | |
| Other | Pierce Streptavidin Magnetic Beads | Thermo Fisher Scientific | 88817 | |
| Other | Novex DYNAL Dynabeads His-tag Isolation and Pulldown | Thermo Fisher Scientific | 10103D | |

## Strains used for protein expression

For protein expression in *E. coli*, BL21-CodonPlus (DE3)-RIPL competent cells (Agilent) were used. Where indicated, BL21 (DE3) competent cells (NEB) were used instead. To express Doa10 in *S. cerevisiae*, we used a doa10 deletion strain derived from BY4741.

## Constructs

All sequences were from *S. cerevisiae*, except for SNARE proteins which were from *rattus norvegicus*.

### Doa10

As full-length DOA10 is toxic for *E. coli* (**Mandart et al., 1994**), the DOA10 sequence was split in two parts and cloned into two separate plasmids, similarly to as described before (**Swanson et al.,**

*2001*). Sequences coding for Doa10 (amino acids (aa) 1–468, Doa10-N) and Doa10 (aa 225–1319), were both cloned into a pRS426-pGal1 plasmid (*Mumberg et al., 1994*) using XhoI/SpeI restriction sites (plasmids #376 and #375, respectively). At the carboxy terminus, both constructs were appended with a tobacco etch virus (TEV) protease cleavage site followed by a streptavidin-binding peptide (SBP) tag (*Keefe et al., 2001*), in which the single lysine was mutated to arginine, and a short sequence for sortase-mediated labeling with fluorescent dyes (*Popp et al., 2009*). The full C-terminal tag for Doa10 had the sequence GSGENLYFQSGGGMDERTTGWRGGHVVEGLAGELE QLRARLEHHPQGQREPLPETGG. A plasmid containing full-length Doa10 was subsequently generated in *S. cerevisiae* (DOA10 deletion strain) by homologous recombination. To do so, plasmid #375 was linearized (starting from Doa10 residue 225) and an N-terminal fragment was generated from plasmid #376 (Doa10 residues 1–257) by PCR. The N-terminal fragment contained 80 to 100 nt overlaps with the linearized plasmid. Both PCR-products were co-transformed into *S. cerevisiae*. Correct homologous recombination was confirmed by sequencing of the PCR-amplified insert after preparation of total DNA of the generated strain (yAST112). The construct for expression of Doa10-C (aa 434–1319) contained an N-terminal SBP-SUMO* tag (*Liu et al., 2008*) and a C-terminal sortase (LPETGG) tag.

## Ubc6

UBC6 from *S. cerevisiae* and its variants were cloned into the K27SUMO vector using the SfoI restriction site (*Stein et al., 2014*). This vector encodes an N-terminal $His_{14}$-SUMO-tag. Ubc6 and its variants were appended with a C-terminal LPETGG tag for sortase-mediated labeling. Expression constructs for Ubc6 and its variants were generated by Gibson assembly (NEB) and site-directed mutagenesis (NEB):

$Ubc6_{SybTM}$ contained the cytosolic Ubc6 domain (aa 1–231) fused to the sequence of the transmembrane (TM) domain of rat synaptobrevin 2 (Syb, aa 96–116). Vice versa, $Syb_{Ubc6TM}$ contained the cytosolic part of Syb (aa 1–95) fused to the Ubc6 TM domain (aa 232–250). Constructs containing a Syb TM domain contained a linker between the TM domain and the LPETGG tag for sortase labeling with the sequence GSGSATGSGGS.

To generate catalytically inactive Ubc6, the active-site Cys (C87) was mutated to Ala ($Ubc6_{C87A}$).

$Ub-Ubc6_{C87A}$ and $Ub-Ubc6_{C87A/SybTM}$ contained $ubiquitin_{V76}$ (aa 1–76) which was inserted between the sequence encoding the $His_{14}$-SUMO tag and the respective Ubc6 variant. For efficient Ulp1-cleavage, a linker sequence (coding for GSG) was inserted between the $His_{14}$-SUMO tag and ubiquitin.

SUMO-Ubc6 contained a C-terminal TEV-cleavage site flanked by linker sequences which was introduced between the Ubc6 TM domain and the LPETGG tag resulting in GSGS-ENLYFQS-SGLPETGG.

Ubc6-SBP contained a C-terminal TEV-cleavage site separated from Ubc6 by a linker (GSGEN-LYFQSGGG) followed by an SBP-tag and residues LPETGG for sortase-mediated labeling. The coding sequence for $His_{14}$-SUMO-Ubc6-SBP was inserted into a pET39b(+) vector (Novagen) right after the DsbA signal sequence.

## Cue1

Cue1 was engineered with an N-terminal $His_{14}$-SUMO-tag and a C-terminal TEV-cleavage site followed by an SBP-tag separated from Cue1 by a linker (resulting in the same C-terminal tag as the one for Ubc6-SBP). A short linker (coding for SGS) was introduced between the $His_{14}$-SUMO tag and Cue1. The coding sequence for this construct was inserted into the pET39b(+) vector (Novagen) right after the DsbA signal sequence. After TEV-cleavage during purification, the sequence for the C-terminal end of Cue1 is GSGENLYFQ.

## Get3

The coding sequence for Get3 (and $Get3_{D57N}$) was inserted into the K27SUMO vector using the SfoI restriction site. The expression construct contained an N-terminal $His_{14}$-SUMO tag.

$Get3_{I193D}$ was expressed from a pET28 vector (kind gift from Blanche Schwappach).

## SNAREs

All constructs for expression of SNAREs have been previously described (*Hernandez et al., 2012*; *Stein et al., 2007*).

| Construct | Plasmid Number |
|---|---|
| Doa10 (aa 1–468, Doa10-N) in pRS426-pGal1 | 376 |
| Doa10 (aa 225–1319) in pRS426-pGal1 | 375 |
| Doa10 (aa 434–1319, Doa10-C) in pRS426-pGal1 | 557 |
| Ubc6 in K27SUMO | 343 |
| $Ubc6_{C87A}$ in K27SUMO | 682 |
| $Ub\text{-}Ubc6_{C87A}$ in K27SUMO | 702 |
| $Syb_{Ubc6TM}$ in K27SUMO | 509 |
| $Ubc6_{SybTM}$ in K27SUMO | 536 |
| $Ub\text{-}Ubc6_{C87A/SybTM}$ in K27SUMO | 815 |
| Ubc6 (incl. C-terminal TEV cleavage site) in K27SUMO | 508 |
| Ubc6-SBP in pET39b(+) | 633 |
| Cue1 in pET39b(+) | 672 |
| Get3 in K27SUMO | 504 |
| Get3 D57N in K27SUMO | 522 |
| Get3 I193D | *Mateja et al., 2009* |
| Syntaxin 1A 183–288, Syb2 49–96 in pETDuet-1 | *Stein et al., 2007* |
| $SNAP25A_{nocys}$ in pET28a | *Fasshauer et al., 1999* |
| Synaptobrevin 2 in pET28a | *Stein et al., 2007* |

## Expression and purification of proteins

For expression of Doa10, yeast cells were grown in synthetic complete medium containing 2% (w/v) Glucose and amino acid drop-out supplements at 30°C (Formedium). An overnight culture (grown for 24 hr) was diluted 1:50 into fresh medium. After 24 hr of growth, expression was induced by addition of yeast extract, peptone and galactose to final concentrations of 1%, 2%, and 2% (w/v), respectively. After 17–19 hr, the cells were harvested at 3000 x g, washed once with $ddH_2O$, resuspended in a minimal amount of $H_2O$ and stored at −80°C.

To prepare cell lysates, 150 g of cells were resuspended in 900 mL of cold $H_2O$ and incubated with 2 mM DTT for 15 min on ice. All subsequent steps were done at 4°C. The cells were pelleted at 3000 x g and resuspended in lysis buffer (20 mM HEPES/KOH pH 7.5, 5 mM potassium acetate, 600 mM mannitol, 0.5 mM EDTA). PMSF (1 mM) and Pepstatin A (2 µM) were added freshly. The cell suspension was then added to a bead beating chamber (total volume 300 mL) filled up to 1/3 with zirkonia beads. Cells were lysed in a Bead Beater (Biospec Products) with cycles of 20 s on and 2 min breaks in between for 50 min. Beads were filtered off, and the lysate was centrifuged at 1500 x g for 10 min. The supernatant was subsequently centrifuged at 40,000 x g for 45 min in a Ti45 rotor. The pelleted crude membrane fraction was resuspended in 200 mL lysis buffer by douncing and again pelleted at 180,000 x g for 30 min. The pellet was resuspended in 40 mL buffer, frozen in liquid nitrogen, and stored at −80°C. The total protein concentration of the membrane fraction was determined using the Pierce660 nm Protein Assay (Thermo Scientific).

To purify SBP-tagged Doa10, the membrane fraction was solubilized with 1.3% (w/v) GDN (Anatrace) at a protein concentration of 3–4 mg/mL in 20 mM HEPES/KOH pH7.4, 300 mM potassium chloride, 0.5 mM TCEP, 5 mM magnesium acetate, supplemented with 1 mM PMSF and 1 Pierce cOmplete EDTA-free protease inhibitor cocktail (Roche) per 100 mL solubilization volume. After 1 hr solubilization, insoluble material was pelleted at 40,000 rpm for 30 min in a Ti45 rotor. The supernatant was added to 4 mL Pierce High Capacity Streptavidin Agarose slurry (Thermo Fisher Scientific) and incubated for 3 hr. The beads were filtered off and washed with 4 × 25 mL of wash buffer (20

mM HEPES/KOH pH 7.4, 150 mM potassium chloride, 5 mM magnesium acetate, 0.5 mM TCEP, 150 μM GDN). Doa10 was eluted with wash buffer supplemented with 2 mM biotin. Doa10 was further purified by sucrose density gradient ultracentrifugation. Gradients were prepared with two solutions where the less dense solution contained GDN (solution A: 20 mM HEPES/KOH pH 7.4, 150 mM potassium chloride, 2 mM magnesium acetate, 10% (w/v) sucrose, 0.5 mM TCEP, 100 μM GDN, solution B: 20 mM HEPES/KOH pH7.4, 150 mM potassium chloride, 2 mM magnesium acetate, 25% (w/v) sucrose, 0.5 mM TCEP). Gradients were prepared using a gradient mixer (Gradient Master, Biocomp Instruments) at RT and kept at 4°C. 500 μL sample was loaded on top of the gradient. After centrifugation at 40,000 rpm for 19 hr in a SW41Ti rotor, the gradient was harvested in 500 μL fractions. Doa10-containing fractions were concentrated with Amicon Ultra Centrifugal Filters (Merck) using a 100 kDa cut-off. The same protocol was used for purification of SBP-tagged Doa10-N and Doa10-C.

For bacterial expression, an overnight culture was diluted 1:50 into Terrific Broth and grown at 37°C. At an $OD_{600}$ of 0.5, the cells were shifted to 18°C and expression was induced with 0.5 mM IPTG. After approximately 20 hr of induction, cells were harvested at 4000 rpm, resuspended in buffer Im30 (50 mM Tris/HCl pH8.0 (at 4°C), 500 mM NaCl, 30 mM Imidazole) and stored at −20°C. Cells were lysed using a microfluidizer (17,000 psi, two passages). Immediately afterwards, 1 mM PMSF was added. Cell debris and unbroken cells were pelleted (1500 x g, 10 min). A membrane fraction was prepared by ultracentrifugation of the supernatant (40,000 rpm, 45 min, Ti45 rotor). The pellet was resuspended in buffer Im30 by douncing, frozen in liquid nitrogen, and stored at −80°C.

Ubc6 and its variants were purified as described (*Vasic et al., 2020*).

To purify Ubc6-SBP, an additional purification step was included to ensure that only full-length Ubc6-SBP was purified. After size-exclusion chromatography, the protein was diluted to 0.5 mg/mL and bound to Pierce High Capacity Streptavidin Agarose (Thermo Scientific). After washing the beads with buffer containing 20 mM HEPES/KOH pH 7.4, 250 mM sodium chloride, 0.2 mM TCEP, 0.03% (w/v) n-dodecyl-β-D-maltopyranoside (DDM, Carl Roth), Ubc6-SBP was eluted with buffer supplemented with 2 mM biotin.

To purify SUMO-Ubc6 (containing a C-terminal TEV cleavage site), protein was eluted from the Ni-NTA resin with buffer containing 50 mM Tris/HCl pH 8.0 (at 4°C), 500 mM sodium chloride, 0.03% (w/v) DDM and 500 mM imidazole, and then further purified by size-exclusion chromatography (Superdex 200), as for Ubc6.

To purify Get3, bacterial lysate was cleared by ultracentrifugation (40,000 rpm, 45 min, 4°C, Ti45 rotor) and the supernatant incubated with Ni-NTA slurry (6 ml slurry for 6 L of culture) for 2 hr. Beads were filtered off and washed with 4 × 50 mL buffer Im30 and 50 mL of buffer Im10 (20 mM Tris/HCl pH8.0 (at 4°C), 200 mM NaCl, 10 mM Imidazole). Get3 was eluted from beads by cleavage with Ulp1 as described above. The elution fractions were supplemented with 1 mM DTT and further purified by size-exclusion chromatography using a Superdex 200 HiLoad16/60 column (GE Healthcare) equilibrated with 20 mM HEPES/KOH pH 7.4, 200 mM NaCl, 1 mM DTT.

Uba1, Ubc7, Cdc48 and Ufd1/Npl4 were purified as described (*Stein et al., 2014*). Cue1 was purified as described (*Vasic et al., 2020*).

To express the t-SNARE complex, plasmids encoding syntaxin-1a (aa 183–288), synaptobrevin-2 (aa 49–96) (pETDuet-1 vector) and SNAP-25A (pET28a vector) were co-transformed into BL21 (DE3) *E. coli* cells (NEB) and expressed as described previously (*Stein et al., 2007*). Briefly, at an $OD_{600}$ of 0.5, the cells were shifted to 18°C and induced with 0.5 mM IPTG. After approximately 20 hr of induction, the cells were harvested at 4000 rpm, resuspended in buffer Im8 (50 mM Tris/HCl pH8.0 at 4°C, 500 mM NaCl, 8 mM Imidazole) and stored at −20°C. After cell lysis using a microfluidizer in the presence of 1 mM PMSF and Complete protease inhibitor, the lysate was cleared by ultracentrifugation (40,000 rpm, 30 min, Ti45 rotor). The pellet was solubilized in buffer Im8 supplemented with 5% (w/v) sodium cholate (Sigma), 2 M urea, 200 mM sucrose and 1 mM PMSF (30 min, RT). Insoluble material was pelleted by ultracentrifugation (40,000 rpm, 30 min, 4°C, Ti45 rotor). HisPur Ni-NTA resin (6 mL for 6 L of culture) was added to the supernatant and incubated for 3 hr at 4°C while rotating. Beads were filtered off and washed with 4 × 50 mL wash buffer (20 mM Tris/HCl pH 8.0 (at 4°C), 500 mM NaCl, 8 mM imidazole, 200 mM sucrose, 2% (w/v) octyl glucoside (OG, Glycon Biochemicals)). Protein was eluted with wash buffer supplemented with 400 mM Imidazole. 1 mM DTT and 0.05 mg/mL of thrombin (100x stock prepared in 50% (w/v) glycerol) were added to the elution fractions and incubated at 4°C overnight. The solution was then diluted to a conductivity of

15 mS/cm with buffer A (20 mM Tris/HCl pH7.4 (RT), 1 mM DTT, 200 mM sucrose, 2% (w/v) OG). The protein was further purified by ion exchange chromatography on a MonoQ column (GE healthcare) equilibrated with 20 mM Tris/HCl pH7.4 (RT), 150 mM NaCl, 1 mM DTT, 200 mM sucrose, 2% (w/v) OG and eluted in a gradient until 450 mM NaCl (elution at ~25 mS/cm).

ATP synthase (from *Bacillus* PS3) was expressed and purified as described previously (*Schenck et al., 2009*; *Suzuki et al., 2002*). Shortly, ATP synthase was purified in the detergent DDM via a His$_{10}$-tag attached to the β subunit. After Ni-NTA affinity chromatography, the dialysis step was omitted and the sample was directly further purified via ion exchange chromatography (MonoQ). Detergent was exchanged to GDN subsequently by size exclusion chromatography (Superose 6 16/60, equilibrated with 10 mM HEPES, 100 mM KCl, 5 mM MgCl$_2$ and 50 μM GDN, pH 7.4). The protein was stored at 4°C for up to two weeks.

To express Syb, a plasmid encoding His$_6$-thrombin-Syb was transformed into BL21 (DE3) *E. coli* cells (NEB). Expression and preparation of a membrane fraction were done as described above, in buffer Im15 (50 mM Tris/HCl pH 8.0, 500 mM NaCl, 15 mM imidazole). The membrane fraction was solubilized in buffer Im15 supplemented with 2.5% (w/v) sodium cholate for 30 min. After ultracentrifugation, solubilized material was incubated with Ni-NTA slurry (6 mL for 6 L culture) for 3 hr. Beads were filtered off and washed with 2 × 50 mL wash buffer Im15 supplemented with 1.5% (w/v) sodium cholate and subsequently with 4 × 50 mL wash buffer Im15 supplemented with 5 mM decylmaltoside (DM, Glycon Biochemicals). Protein was eluted with buffer Im15 containing 400 mM imidazole and 5 mM DM. The solution was dialyzed overnight against 10 mM MOPS, 50 mM NaCl, 1 mM DTT, 1 mM EDTA pH 7.0 (10 kDa MWCO), in the presence of 0.05 mg/mL thrombin. The protein was further purified by ion exchange chromatography on a MonoS column (GE healthcare) equilibrated with 10 mM MOPS, 50 mM NaCl, 1 mM EDTA and 1 mM DTT pH 7.0 and eluted in a salt gradient to 500 mM NaCl.

## Sortase-mediated labeling

Proteins were labeled at their C-terminal LPETGG tag with the previously described technique sortase-mediated transpeptidation (*Popp et al., 2009*). A peptide with the sequence GGGC was labeled at its cysteine residue with a maleimide dye. Peptide dissolved in 100 mM HEPES/KOH pH 7.4 was added to dye (DyLight680 maleimide or DyLight800 maleimide, Thermo Scientific) in 1.5-fold molar excess. After labeling at RT for 2 hr, the reaction was stopped with 10 mM DTT. To label peptide with AlexaFluor 488 C5 Maleimide (Thermo Scientific), peptide and dye were both dissolved in 100 mM HEPES/KOH pH7.4 and then mixed in a 1:1 molar ratio. A pentamutant P94R/D160N/D165A/K190E/K196T of SrtA from *S. aureus* was purified from *E. coli* (*Chen et al., 2011*).To label proteins, 3-fold molar excess of labeled peptide, 10 mM CaCl$_2$ and SortA were added to the protein. SortA was added to 1/7 of the total concentration of reactants (peptide and protein). After labeling for 16–20 hr at 4°C, the reaction was separated by size-exclusion chromatography.

## Reconstitution into proteoliposomes

### Preparation of protein-free liposomes

The following lipids were purchased from Avanti Polar Lipids: 16:0-18:1 PC (POPC, 1-palmitoyl-2-oleoyl-glycero3-phosphocholine), 18:1 (Δ9-Cis) PE (DOPE, 1,2-dioleoyl-sn-glycero-3-phosphoethanolamine), 18:1 PS (DOPS, 1,2,-dioleoyl-sn-glycero-3-phospho-L-serine), 18:1 Biotinyl PE (Biotinyl-PE, 1,2-dioleoyl-sn-glycero-3-phosphoethanolamine-N-(biotinyl)), 18:1 Liss Rhod PE (Rhd-PE, 1,2-dioleoyl-sn-glycero-3-phosphoethanolamine-N-(lissamine rhodamine B sulfonyl)). Ergosterol (≥95%, HPLC) was purchased from Sigma-Aldrich.

Large unilamellar liposomes were prepared by reverse-phase evaporation as described (*Hernandez et al., 2012*). Briefly, lipids were dissolved in chloroform and mixed at a molar ratio of 60:20:10:10 (POPC: DOPE: DOPS: Ergosterol). Chloroform was subsequently removed using a rotary evaporator by lowering the pressure step-wise to 20mbar. The lipid film was then dissolved in 1 mL diethyl ether (when preparing 1 mL of liposomes with a final concentration of 20 mM lipid) and 300 μL of buffer L (20 mM HEPES/KOH pH7.4, 150 mM potassium chloride, 5 mM magnesium acetate) was added. The sample was sonicated for 1 min on ice (Branson Sonifier 450, 100% duty cycle, microtip limit 1). Afterwards the ether was removed at 500 mbar. After 10 min, 700 μL of buffer L was added and the pressure was gradually decreased to 100 mbar until diethyl ether was completely

removed. The volume was adjusted to 1 mL with ddH$_2$O. The resulting lipid suspension was extruded through a polycarbonate filter (11 x through a 0.4 µM filter, 21x through a 0.1 µM filter) using the Mini extruder kit (Avanti Polar Lipids). Protein-free liposomes were used for up to 2 weeks after preparation. For pulldown experiments via co-reconstituted biotinylated lipids, lipids were used in a molar ratio of 57.5:20:10:10:0.5:2 (POPC: DOPE: DOPS: Ergosterol: Rhd-PE: Biotinyl-PE).

## Reconstitution of proteins into liposomes

To reconstitute proteins into liposomes, protein-free liposomes were mixed with detergent and proteins and subsequently incubated for 1 hr at RT prior to detergent removal. The detergent concentration used for solubilization can be described by the R-value (*Rigaud and Lévy, 2003*). The R-value is defined as the ratio of the total detergent concentration above the critical micellar concentration and the total lipid concentration (R = [D$_{total}$ – D$_{CMC}$]/[lipid]).

To co-reconstitute Ubc6 and t-SNARE, protein-free liposomes (4 mM final lipid concentration) were mixed with OG (R-value of 2), proteins and buffer D (buffer L supplemented with 1 mM DTT). t-SNARE and Ubc6 were reconstituted at a molar lipid: protein ratio of 1000 and 2000, respectively. After incubation for 1 hr at RT, the detergent was removed by dialysis against a 1000 x volume of buffer D at RT in two steps using dialysis cassettes (16 hr with 2000 kDa cut-off, 2 hr with 10,000 kDa cut-off; Slide-a-Lyzer from Thermo Scientific). Biobeads (SM-2 resin, Bio-Rad) were added to the buffer to bind OG (2 g/L). When liposomes contained Ubc6$_{SybTM}$, Ubc6 and t-SNARE, both Ubc6 variants were reconstituted at a molar lipid: protein ratio of 2000.

To reconstitute Doa10 and Syb, protein-free liposomes (4 mM final lipid concentration) were mixed with DM (R-value of 0.55), proteins and buffer T (buffer L supplemented with 0.1 mM TCEP). Doa10 and Syb were reconstituted at a molar lipid: protein ratio of 5000 and 2000, respectively. For ATP synthase, also a molar lipid: protein ratio of 5000 was used. For ubiquitination experiments, Cue1 was co-reconstituted with Doa10 and Syb at a molar lipid: protein ratio of 20,000. After incubation for 1 hr at RT, the detergent was removed by incubation with resin from Pierce detergent removal spin columns (Thermo Scientific) in three subsequent steps (45 mg washed resin to 130 µL reconstitution mix in each step). Resin incubation was performed while rotating the sample, at RT for 20 min each. Doa10 truncations were reconstituted at the same lipid: protein ratio, also when both Doa10 truncations were co-reconstituted for the rescue experiment in *Figure 5E,F* and *Figure 5—figure supplement 2*.

After reconstitution into separate liposomes, Ubc6 and Doa10 were subsequently co-reconstituted by SNARE-mediated fusion. Both liposomes sets were therefore mixed by diluting them 1:10 into buffer T (unless otherwise indicated) and incubated for 1 hr at 30°C. To inhibit fusion, t-SNARE liposomes were preincubated with 7-fold excess of a soluble Syb fragment (aa 1–95, Syb$_{sol}$) for 5 min at RT prior to addition of Syb liposomes.

For the experiment in *Figure 4*, D and E, Doa10 was co-reconstituted with either Ubc6$_{SybTM}$, Syb-Ubc6TM, or Ubc6 directly in a 1-step protocol. For this protocol, Doa10 purified in DMNG (f.c. 0.5 mM) was used. Protein-free liposomes (10 mM final lipid concentration) were mixed with DMNG (R-value of 1.5), proteins and buffer T. Ubc6 and Doa10 were both reconstituted at a molar lipid: protein ratio of 10,000. After incubation for 1 hr at RT, the detergent was removed by incubation with Pierce detergent removal spin columns that were pre-washed with buffer T in three subsequent steps (one spin column for 100 µL reconstitution mix in each step). Incubation was performed at RT for 10, 20 and 30 min and the sample eluted by centrifugation at 3,500 rpm for 2 min in a table top centrifuge. To reconstitute Ubc6 or its variants alone (without Doa10), protein-free liposomes (10 mM final lipid concentration) were mixed with OG (R-value of 2.0), proteins and buffer T. Ubc6 was reconstituted at a molar lipid: protein ratio of 10,000. After incubation for 1 hr at RT, the detergent was removed by adding resin from Pierce detergent removal spin columns in three subsequent steps (40, 60, 60 mg resin to 160 µL reconstitution mix in step 1, 2 and 3, respectively). Resin incubation was performed while rotating the sample, at RT for 20 min each and the sample eluted by centrifugation at 3,500 rpm for 2 min in a table top centrifuge.

## Flotation of liposomes

To test for reconstitution of proteins, liposomes were floated in a Nycodenz step gradient. Nycodenz stocks were prepared in buffer L. 50 µL of liposomes were mixed with 50 µL of 80% (w/v)

Nycodenz and overlaid with 40 µL of 30% and 15% (w/v) Nycodenz and 40 µL of buffer L. The gradients were ultracentrifuged at 50,000 rpm for 1 hr at 4°C (S55-S rotor). The gradient was disassembled in six fractions, starting from the top of the gradient. Fractions were analyzed by SDS-PAGE.

## Protease protection

To check the orientation of Ubc6 reconstituted into liposomes, trypsin protease was used. Liposomes were diluted (1:10 in buffer D) and incubated with 6.6 µg/mL trypsin (Roche) at RT. The detergent control contained in addition 1% Triton-X100 (TX100, Anatrace, Anapoe-X-100). The reaction was stopped with 4 mM PMSF and samples were analyzed by SDS-PAGE.

The orientation of Doa10$_{TEV-SBP}$ or Doa10-N$_{TEV-SBP}$ in liposomes was determined by assessing the accessibility of the C-terminal TEV-cleavage site to TEV-protease. Liposomes were diluted 1:10 into buffer D and incubated with 10 µM TEV-protease at RT. The detergent control contained in addition 1% Triton-X100. The reaction was stopped by addition of SDS-sample buffer and samples were analyzed by SDS-PAGE and subsequent Western blotting against the SBP-tag. To determine the orientation of $_{SBP-SUMO*-}$Doa10-C, the accessibility of the N-terminal SBP-SUMO* to Ulp1* protease was assessed (protocol as described above for TEV-protease).

The reconstitution quality of Ubc6 was assessed with a Ubc6 construct containing an N-terminal SUMO tag and a C-terminal TEV-cleavage site followed by a fluorescent dye (SUMO-Ubc6). Liposomes containing SUMO-Ubc6 and t-SNARE were diluted into buffer T 1:20 (f.c. 0.1 µM Ubc6). Ulp1 and/or TEV-protease were added to a f.c. of 10 µM each.

## Pulldowns

For pulldown experiments via the SBP-tag of Doa10, 20 µL of the fusion reaction (supplemented with 0.25 mg/mL bovine serum albumin (BSA)) were incubated with 20 µL of Pierce Streptavidin Magnetic Beads (Thermo Scientific) prewashed with buffer B (buffer T supplemented with 0.25 mg/mL BSA). After binding for 1 hr (rotating, RT), the supernatant of the binding reaction was taken off, the beads washed three times with 100 µL of buffer B and bound proteins eluted with 20 µL of buffer B supplemented with 2 mM biotin. Samples from input, supernatant and elution fractions were analyzed by SDS-PAGE.

For pulldowns via the His-tag of ATP synthase, the fusion reaction was supplemented with 200 mM imidazole and 0.4 mg/mL BSA, and incubated with 20 µL magnetic Dynabeads (His-tag Isolation and Pulldown, ThermoFisher Scientific) (f.c. of 0.2 µM Ubc6, 80 nM ATP synthase). After binding for 30 min (rotating, RT), the supernatant was removed. Samples of input and supernatant were analyzed by SDS-PAGE and immunoblotting for His-tagged β-subunit of ATP Synthase.

## Assays for release by Doa10

### Get3 capture assay

Liposomes were prepared with protein-free liposomes containing 2 mol% biotinyl-PE and 0.5 mol% Rhd-PE and fused as described above. The fusion reaction was diluted 1:2 into buffer T (f.c. Ubc6 = 0.1 µM) and incubated with an excess of Get3 (f.c. 10 µM). After incubation at RT for 16 hr, the reaction was diluted to a f.c. of lipid of 0.2 mM (1:2 dilution) and 0.25 mg/mL BSA was added. The diluted mix was then added to an equal volume of Pierce Streptavidin Magnetic beads (Thermo Scientific, prewashed with buffer B). After binding for 1 hr, the supernatant was removed. Input and supernatant samples were analyzed by SDS-PAGE, and the Rhodamine fluorescence measured as described below.

For the turbidity assay, Ubc6 (in 0.03% (w/v) DDM) was diluted 1:25 into buffer L (f.c. of 1.8 µM Ubc6) in the presence or absence of Get3 (f.c. 1.8 µM or 3.6 µM). The optical density at 360 nm was measured using a UV-2401PC spectrophotometer (Shimadzu Corporation).

### Protease protection assay

For this experiment, both sets of liposomes (SUMO-Ubc6, t-SNARE and Syb liposomes containing no, full-length or truncated Doa10 versions) were diluted 1:5 for the fusion reaction. First, Ulp1 cleavage was performed (f.c. 2 µM Ulp1, 0.2 µM Ubc6, 0.08 µM Doa10). For the subsequent TEV-cleavage, the Ulp1-cleaved sample was diluted 1:2 and incubated with 10 µM TEV-protease. During TEV-

cleavage, 0.5 mM DTT was present. Detergent controls contained 1% TX100. Reactions were stopped by adding SDS sample buffer.

### Antibody accessibility assay

AlexaFluor 488 fluorescence was measured in a Tecan Genios Pro microplate reader using 495/10 nm and 535/25 nm for excitation and emission, respectively. The fluorescence of 30 µL of the fusion reaction was measured in a 96-well plate (Corning, REF 3686) with a f.c. of Ubc6 (labeled with A488) of 0.2 µM. After the signal was stable, the measurement was stopped, anti-Alexa Fluor 488 polyclonal antibody (Invitrogen, #A-11094) was added (diluted 1:15) and the measurement started again. After approx. 40 min, 1 µL TX100 (f.c. 1%) was added to solubilize the liposomes. To analyze the fluorescence traces, the three measurements (equilibration, antibody and detergent addition) were merged. The background (stabilized A488 signal after detergent addition) was subtracted from all measurements. The fluorescence traces were subsequently normalized to the average signal of the last 10 timepoints before antibody addition. To quantify the fraction of released Ubc6, the difference between the normalized values of the samples with and without Doa10 30 min after antibody addition was calculated.

To test for the release of the Ubc6/Syb chimera, liposomes containing Doa10 directly co-reconstituted with A488-labeled Ubc6, Ubc6$_{SybTM}$ or Syb$_{Ubc6TM}$ were first subjected to a pulldown via the SBP-tag of Doa10. Liposomes were diluted 1:8 (f.c. Ubc6 0.125 µM, 0.25 mg/mL BSA) and 50 µL of diluted liposomes were added to 50 µL of Pierce Streptavidin Magnetic Beads (Thermo Scientific) prewashed with buffer B. For the controls, Ubc6-only liposomes were also diluted 1:8 (in the presence or absence of 1:8 diluted Doa10-liposomes) and incubated with beads. After binding for 1 hr (rotating, RT), the supernatant of the binding reaction was taken off, the beads washed three times with 200 µL of buffer B and bound Doa10-liposomes eluted with 40 µL of buffer B supplemented with 2 mM biotin. Samples from input, supernatant and elution fractions were analyzed by SDS-PAGE. 30 µL of eluted fractions (Doa10-containing liposomes) were then added into a 96-well plate. Liposomes lacking Doa10 (containing Ubc6, Ubc6$_{SybTM}$ or Syb$_{Ubc6TM}$) were diluted 1:15 in buffer T. The antibody quenching assay was then carried out as described above.

## Ubiquitination assays

All ubiquitination reactions were performed at 30°C in a thermocycler. The fusion reaction was diluted 1:2 (f.c. of 0.1 µM Ubc6, 0.01 µM Cue1 and 0.04 µM Doa10). The following components were used at the indicated concentrations unless stated otherwise: 0.1 µM Uba1 (E1), 1 µM Ubc7, 120 µM ubiquitin (from *S. cerevisiae*, R and D Systems) and 2.5 mM ATP. All reactions contained 0.1 mg/mL BSA. The ubiquitin mutant K0 (Lifesensors) is derived from human ubiquitin. Reactions were stopped by adding reducing SDS-sample buffer and samples were analyzed by SDS-PAGE.

### Analysis of ubiquitination reactions

To analyze the fraction of non-modified protein, the non-ubiquitinated band was quantified and normalized to the 0 min timepoint. To analyze the ubiquitin chain profile, the fluorescence intensity was quantified along a vertical axis starting from the top of the gel using the line scan function in ImageJ (*Figure 2B* and *Figure 5B*). When different Ubc6-variants were compared (Ubc6 vs Ubc6$_{SybTM}$), the line scan values were normalized to the integral of the whole scan (*Figure 5B*), to account for different sortase labeling efficiencies. To quantify the kinetics for generation of mono-, di-, tri- and tetraubiquitinated species, each band corresponding to one, two, three and four ubiquitins was quantified for every time point and normalized to the non-ubiquitinated band at the 0 min timepoint (*Figure 5—figure supplement 1D*). To calculate the total number of ubiquitins transferred relative to total Ubc6, the values for species modified with 1 to 4 ubiquitins obtained as above were summed up for each timepoint (*Figure 5D* and *Figure 5—figure supplement 2C*).

## Measuring extraction by the Cdc48-complex

Proteoliposomes were prepared with protein-free liposomes containing 2 mol% biotinyl-PE and 0.5 mol% Rhd-PE. Fusion and ubiquitination was carried out as described above. To immobilize liposomes after ubiquitination, the ubiquitination reaction was diluted 1:2 to a final lipid concentration of 0.2 mM total lipid (f.c. of 0.05 µM Ubc6) and BSA was added to a f.c. of 0.25 mg/ml. The diluted

mix was then added to an equal volume of Pierce Streptavidin Magnetic Beads (Thermo Scientific, prewashed with buffer B). After incubation for 1 hr at RT (rotating), the unbound fraction was removed and the beads were subsequently washed 3 x with buffer B. The beads were then resuspended in the same volume of buffer B and 30 µL of the suspension aliquoted in a PCR-strip. The buffer was removed and the beads resuspended in 1 x extraction mixes or 1 x SDS sample buffer. 1 x extraction mixes contained 0.25 mg/mL BSA and where indicated 0.1 µM Cdc48 (hexamer) and 0.1 µM Ufd1/Npl4. Beads were incubated for 30 min at 30°C. The supernatant was removed (containing extracted and soluble proteins). After washing the beads 3 x with buffer B, the bound proteins were eluted by adding 30 µL of 1x SDS sample buffer. Samples of the supernatant and the elution fractions were analyzed by SDS-PAGE. To quantify the liposome immobilization efficiency, the Rhodamine fluorescence was measured in a Tecan Genios Pro microplate reader using 550/10 nm and 590/20 nm for excitation and emission, respectively. To quantify the protein immobilization efficiency, the DyLight680 (Ubc6/Ub-Ubc6$_{C87A}$) fluorescence was measured using the Odyssey scanner (384-well plate, transparent bottom).

## Analysis of extraction reactions

To quantify the extraction efficiency of Ub-Ubc6$_{C87A}$ relative to its ubiquitination status (*Figure 3— figure supplement 1G*), bands corresponding to Ub-Ubc6$_{C87A}$ modified with 1 to 10 ubiquitins were quantified separately and normalized to the corresponding band of the input sample (beads treated with sample buffer).

To quantify the fraction of Ubc6 in the supernatant (*Figure 3C,D*), the fluorescence intensity for Ubc6 modified with 0–5 ubiquitins and for Ubc6 modified with more than five ubiquitins was quantified by drawing a single rectangular box around the respective area using Image Studio and subsequently normalized to the Input (beads treated with 1 x SDS sample buffer). For Ub-Ubc6$_{C87A}$, the unmodified band was counted as monoubiquitinated.

## Experiments with Ubc6-SBP

To reconstitute Ubc6-SBP into proteoliposomes, Ubc6-SBP was preincubated with a 1.25-fold molar excess of tetrameric streptavidin (NEB) for 15 min at RT in the presence of 0.03% DDM to allow for complex formation. The reconstitution conditions were otherwise the same as for the co-reconstitution of Ubc6 and t-SNARE (with 2 µM Ubc6-SBP and 2.5 µM Streptavidin).

To assess the orientation of Ubc6-SBP in liposomes (*Figure 6—figure supplement 1B*), a TEV protease protection assay was carried out as described above, except that liposomes were diluted 1:5.

A biotinylated nanobody was used for the biotinylated protein control. To test if the biotinylated nanobody (anti-GFP, construct for expression kindly provided by Dirk Görlich; purified and biotinylated essentially as described in *Pleiner et al., 2018* is completely biotinylated (*Figure 6—figure supplement 1C*), biotinylated nanobody was supplemented with 0.25 mg/mL BSA and incubated with magnetic streptavidin beads that were prewashed with buffer B or prewashed with buffer B supplemented with 10 mM biotin. After binding for 45 min at RT, samples of input and supernatant fractions were analyzed by SDS-PAGE and stain-free scanning using a GelDoc EZ Imager.

For the antibody accessibility assay, the fusion reaction was first incubated with 1.5 µM biotin or biotinylated protein for 10 min at RT, before the fluorescence measurements were started as described above. A 6-fold molar excess of biotin/biotinylated protein over streptavidin was used. Fluorescence traces were processed and analyzed as described above.

To check if biotinylated protein is capable of releasing Streptavidin from Ubc6-SBP, a flotation assay was used. Liposomes containing Ubc6-SBP and t-SNARE were incubated with Streptavidin for 5 min at RT (f.c. 0.9 µM Streptavidin, 1.4 µM Ubc6-SBP). A 10-fold excess of biotin or biotinylated nanobody was then added and after incubation for another 5 min, a sucrose density gradient (40% (w/v), 30% (w/v), 15% (w/v) and buffer T layer) was assembled. Flotation was carried out as described above. Samples were analyzed by SDS-PAGE and stain-free scanning using a GelDoc EZ Imager. Streptavidin and Ubc6-SBP levels were quantified using ImageJ. Intensity values for Streptavidin were normalized to Ubc6-SBP levels and subsequently to the buffer control.

To measure extraction of Ubc6-SBP by the Cdc48-complex using the antibody accessibility assay, fusion and ubiquitination reactions were carried out as described above, except that liposomes were

diluted 1:5 instead of 1:10 for the fusion reaction. After 30 min of ubiquitination (total volume of 30 μl with f.c. 0.2 μM Ubc6-SBP, 0.08 μM Doa10, 0.02 μM Cue1, 0.1 μM Uba1, 1 μM Ubc7, 120 μM ubiquitin and 2.5 mM ATP), additional 2.5 mM ATP were added and the fluorescence measurement was started (plate reader preheated to 30°C). After 10 min, anti-A488 antibody (1:15 diluted) and 3 μL of 10 x Cdc48/UN mix were added. 10 x Cdc48/UN mix contained 2 μM Cdc48 (hexamer), 2 μM Ufd1/Npl4 as well as 1 mM ATP. To quantify the fraction of extracted Ubc6-SBP, fluorescence traces were processed as described above and the difference between the normalized values of samples with and without Cdc48/UN 30 min after antibody addition was calculated.

To test, if streptavidin stays in the liposome lumen during extraction (*Figure 6E,F*), samples were floated after the extraction assay in a Nycodenz step gradient as described above. The samples were prepared with the following modifications: To ensure sufficient detection levels, the liposomes were diluted 1:4 for the fusion reaction. The ubiquitination reaction (f.c. 0.3 μM Ubc6, 0.12 μM Doa10, 0.03 μM Cue1, 0.1 μM Uba1, 1 μM Ubc7, 120 μM ubiquitin and 2.5 mM ATP) was carried out as described above. After 30 min of ubiquitination, the extraction assay was performed in the plate reader as described above, in the presence of 0.3 μM Cdc48/UN complex (f.c. Ubc6 = 0.25 μM). After 30 min, 2 mM biotin was added to 50 μL sample, a Nycodenz step gradient was assembled and the flotation carried out as described above. Samples were analyzed by SDS-PAGE and stain-free scanning using a GelDoc EZ Imager. Streptavidin levels were quantified using ImageJ. Intensity values were normalized to the sample without Cdc48/UN.

## Analysis

Samples were mixed with SDS sample buffer (stock used as 3 x contained 12% (w/v) SDS, 30% (w/v) glycerol, 0.05% Coomassie blue G-250, 150 mM Tris/HCl pH 7.0% and 6% (v/v) ß-mercaptoethanol for reducing sample buffer (*Schägger, 2006*). Samples were heated at 70°C (Streptavidin-containing samples were boiled) and analyzed by SDS-PAGE using CRITERION TGX stain-free precast gels (Bio-Rad).

Fluorescent proteins were detected using an Odyssey scanner (Li-Cor) for DyLight680 and DyLight800-labeled proteins, and an FLA-700 fluorescence scanner (Fujifilm) for AlexaFluor488-labeled proteins. To detect streptavidin, samples were run on CRITERION TGX stain-free precast gels (Bio-Rad) and scanned with a GelDoc EZ Imager (Bio-Rad). Colloidal Coomassie staining was used (*Dyballa and Metzger, 2009*). SBP-tagged proteins were analyzed by western blotting where indicated. After transfer on a nitrocellulose membrane using the Trans-Blot Turbo Transfer System (Bio-Rad), the membrane was blocked with 5% skim milk powder (dissolved in TBS-T) for 1 hr at RT. A 1:2500 dilution of anti-SBP antibody (clone 20, mouse monoclonal, MAB10764, Millipore), and a 1:15,000 dilution of secondary antibody (goat anti-mouse, IRDye 800 CW or IRDye 680RD) were used for detection. For the analysis of His-tagged ATP synthase, the nitrocellulose membrane was blocked with 2% (w/v) BSA (dissolved in PBS containing 0.1% (v/v) Tween and 0.1% (v/v) Triton-X-100). A 1:500 dilution of anti-His$_6$ tag antibody (Dia-900, Clone13/45/31-2, mouse monoclonal, Dianova), and a 1:15,000 dilution of secondary antibody (goat anti-mouse, IRDye 800 CW) were used for detection. Antibodies were diluted in the respective blocking buffer. Gels were quantified using ImageStudio Lite (Li-Cor). Fiji (ImageJ) was used for quantification of ubiquitin chain profiles (plot profile function) as well as streptavidin (gel analyzer function) (*Schindelin et al., 2012*).

## Acknowledgements

We thank Blanche Schwappach (UM Göttingen, Germany) for providing Get3 constructs, Iris Bickmeyer and Nupur Nupur for technical assistance, and Marina Rodnina, Alex Faesen, Tom Rapoport and Blanche Schwappach for comments on the manuscript. This work was supported by the European Research Council (ERC) under the Horizon2020 research and innovation program (grant # 677770), by the Deutsche Forschungsgemeinschaft SFB1190, TP15 (both to AS), and the Boehringer Ingelheim Fonds (to VV).

## Additional information

### Funding

| Funder | Grant reference number | Author |
| --- | --- | --- |
| H2020 European Research Council | 677770 | Alexander Stein |
| Deutsche Forschungsge-meinschaft | SFB1190, TP15 | Alexander Stein |

The funders had no role in study design, data collection and interpretation, or the decision to submit the work for publication.

### Author contributions
Claudia C Schmidt, Conceptualization, Data curation, Formal analysis, Validation, Investigation, Visualization, Methodology, Writing - original draft, Writing - review and editing; Vedran Vasic, Methodology; Alexander Stein, Conceptualization, Supervision, Funding acquisition, Methodology, Writing - original draft, Project administration, Writing - review and editing

### Author ORCIDs
Claudia C Schmidt ⓘ https://orcid.org/0000-0002-7642-7081
Vedran Vasic ⓘ https://orcid.org/0000-0003-2575-1006
Alexander Stein ⓘ https://orcid.org/0000-0002-2696-0611

### Decision letter and Author response
Decision letter https://doi.org/10.7554/eLife.56945.sa1
Author response https://doi.org/10.7554/eLife.56945.sa2

## Additional files

### Supplementary files
• Transparent reporting form

### Data availability
All data generated or analysed during this study are included in the manuscript and supporting files.

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
