## [Decision Letter]

**Acceptance summary:**

This study investigates how a membrane-embedded protein is dislocated from the endoplasmic reticulum as part of its degradation pathway. By reconstituting the dislocation process with a set of purified factors implicated in this degradation pathway from previous genetic studies, this study clarifies the precise role of each factor and provides a mechanistic model for how dislocation is achieved. The availability of a purified reconstituted system is an important step forward in the field because it provides a solid foundation for many future mechanistic and structural studies. The pathways of degradation from the endoplasmic reticulum are of wide importance because they are crucial for maintaining homeostasis in the secretory pathway of all eukaryotic cells.

**Decision letter after peer review:**

Thank you for submitting your article "Doa10 is a membrane protein retrotranslocase in ER associated protein degradation" for consideration by *eLife*. Your article has been reviewed by three peer reviewers, one of whom is a member of our Board of Reviewing Editors, and the evaluation has been overseen by David Ron as the Senior Editor. The following individual involved in review of your submission has agreed to reveal their identity: Yihong Ye (Reviewer #2).

The reviewers have discussed the reviews with one another and the Reviewing Editor has drafted this decision to help you prepare a revised submission.

Summary:

Endoplasmic reticulum-associated protein degradation (ERAD) is a conserved protein quality control pathway that helps to maintain protein homeostasis in the secretory system in eukaryotic cells. During this process, misfolded ER lumenal or membrane proteins are retrotranslocated into the cytosol where they are degraded by the proteasome. Previously studies have highlighted the importance of two highly conserved ER-membrane associated ubiquitin ligases, Hrd1 and Doa10 in ERAD in budding yeast. A large body of work suggests that Hrd1 acts as a protein translocase that facilitates the movement of misfolded lumenal and membrane ER proteins into the cytosol for degradation. Comparatively little mechanistic information is available about how Doa10 functions in ERAD. The manuscript by Schmidt et al. makes a major contribution toward this goal by reconstituting Doa10-mediated dislocation of a membrane protein in vitro using purified components. The study provide strong evidence that Doa10 can facilitate passive dislocation of a single transmembrane segment (from the substrate Ubc6) to the cytosol when there is a “trap” to capture it. In the absence of such a trap, the authors show that polyubiquitin on the substrate is used by the cytosolic Cdc48 complex to dislocate the substrate in an ATP-dependent reaction. Finally, the authors provide evidence that the Cdc48-mediated reaction is sufficient to unfold lumenal domains to facilitate their translocation to the cytosol. The study was elegantly designed and the data presented are convincing. The findings represent an important step forward in the field and provide a solid foundation for many future mechanistic studies.

Revisions:

The reviewers generally agreed that the study is convincing with the need of only some clarifications, additional explanations, some possible caveats, and some additional discussion. The key major points to be discussed or clarified are as follows:

1) Two of the reviewers wondered whether a control multi-pass membrane protein (other than Doa10) would similarly allow passive dislocation of Ubc6, perhaps due to some non-specific perturbation of the membrane or artifact of the reconstitution. At issue is whether the release of Ubc6 is highly specific to Doa10. Discussion among the reviewers concluded that non-specific dislocation by any protein seems pretty unlikely, but in the absence of an experiment comparing the dislocation reaction in liposomes containing Doa10 versus another similarly sized multi-pass membrane protein, this possibility should be acknowledged in the text.

2) The relationship between the passive Doa10-mediated dislocation described here and the role of Cdc48 and ubiquitination should be discussed more thoroughly. Genetic experiments indicate that ubiquitination and Cdc48 are strictly required for dislocation, but the authors nicely show that one can get dislocation without these steps. The authors should therefore place their observations into the context of these earlier studies and explain their thoughts about whether in certain contexts, a membrane protein like Ubc6 can be dislocated without the need for ubiquitination or Cdc48.

3) The proportion of ubiquitinated protein that is released is relatively small compared to passive release (which is almost complete). This is presumably because there is no Get3 in that experiment. This should be made explicit in the text or legend.

4) The authors should consider re-wording the statements about unfolding because the experiment didn't really measure this, instead using the dissociation of SBP from streptavidin as a surrogate. This surrogate is quite convincing because the bonds that need to be broken to dissociate Streptavidin and SBP are comparable to the types of intramolecular interactions that keep a protein folded. Thus, if a force from the cytosol can break the SBP interaction, it is easy to imagine unfolding working the same way. This chain of logic should be more explicit.

---

## [Author Response]

Revisions:The reviewers generally agreed that the study is convincing with the need of only some clarifications, additional explanations, some possible caveats, and some additional discussion. The key major points to be discussed or clarified are as follows:1) Two of the reviewers wondered whether a control multi-pass membrane protein (other than Doa10) would similarly allow passive dislocation of Ubc6, perhaps due to some non-specific perturbation of the membrane or artifact of the reconstitution. At issue is whether the release of Ubc6 is highly specific to Doa10. Discussion among the reviewers concluded that non-specific dislocation by any protein seems pretty unlikely, but in the absence of an experiment comparing the dislocation reaction in liposomes containing Doa10 versus another similarly sized multi-pass membrane protein, this possibility should be acknowledged in the text.

We agree that a non-related membrane protein would be a useful control for our experiments to exclude the possibility that the observed retrotranslocation of Ubc6 in the presence of Doa10 is due to non-specific perturbations of the lipid bilayer in the presence of a large multipass membrane protein. We have therefore performed experiments with ATP synthase as a control protein. This protein contains 20 TM segments and thus has a comparable TM domain as Doa10 (Guo et al., 2019). Using the antibody accessibility assay, we show that presence of ATP synthase does not lead to retrotranslocation of Ubc6 (Figure 1—figure supplement 3). This figure also shows the characterization of ATP synthase liposomes. Of note, the same detergents and reconstitution protocol were used to reconstitute ATP synthase as for Doa10. Thus, we conclude that retrotranslocation of Ubc6 is not due to some non-specific perturbation of the membrane caused by any multipass TM protein (Results paragraph six).

2) The relationship between the passive Doa10-mediated dislocation described here and the role of Cdc48 and ubiquitination should be discussed more thoroughly. Genetic experiments indicate that ubiquitination and Cdc48 are strictly required for dislocation, but the authors nicely show that one can get dislocation without these steps. The authors should therefore place their observations into the context of these earlier studies and explain their thoughts about whether in certain contexts, a membrane protein like Ubc6 can be dislocated without the need for ubiquitination or Cdc48.

Genetic and biochemical experiments indicate that for proteasomal degradation of membrane protein substrates of Doa10, including Ubc6, (same applies to other ligases), Cdc48 is strictly required (e.g. (Ravid et al., 2006)). In the case of the multi-spanning Doa10 substrate Ste6, it was also shown in fractionation experiments that retrotranslocation requires ubiquitination and Cdc48 action (Nakatsukasa et al., 2008; Nakatsukasa and Kamura, 2016). However, our experiments indeed raise the possibility that a mildly hydrophobic substrate such as Ubc6 is retrotranslocated in a Doa10-dependent, but ubiquitination- and Cdc48-independent manner in the cell. Such a cytosolic pool of Ubc6 would probably have a non-degradative fate and be subject to re-insertion. Experimentally, it would be indistinguishable from a Ubc6 pool that has not been inserted into the ER yet (the use of an N-glycosylation tag which leaves a scar after deglycosylation in the cytosol might be an experimental solution to this problem). Further studies are required to test if such a process actually occurs in appreciable amounts.

We now discuss this point in paragraph four of the Discussion. In this section we also discuss the role of chaperones in the degradation of membrane proteins and the influence of hydrophobicity on passive retrotranslocation, as suggested.

3) The proportion of ubiquitinated protein that is released is relatively small compared to passive release (which is almost complete). This is presumably because there is no Get3 in that experiment. This should be made explicit in the text or legend.

We agree that a missing factor might be the reason for why the efficiency in the Cdc48 experiment in Figure 3 is limited and only approximately 40-50 % of ubiquitinated Ubc6 is extracted. Chaperones or accessory factors such as Ubx2 might increase the efficiency of extraction. We have acknowledged this inefficiency in the text (Results paragraph eight).

4) The authors should consider re-wording the statements about unfolding because the experiment didn't really measure this, instead using the dissociation of SBP from streptavidin as a surrogate. This surrogate is quite convincing because the bonds that need to be broken to dissociate Streptavidin and SBP are comparable to the types of intramolecular interactions that keep a protein folded. Thus, if a force from the cytosol can break the SBP interaction, it is easy to imagine unfolding working the same way. This chain of logic should be more explicit.

Thank you for pointing this out. We have re-worded our statement according to your suggestion and explained why the dissociation of streptavidin from SBP can be interpreted as protein unfolding (Discussion section).